# Representing Farmer Irrigated Crop Area Adaptation in a Large-Scale Hydrological Model

Jim Yoon[1], Nathalie Voisin[1], Christian Klassert[2], Travis Thurber[1], Wenwei Xu[1]

[1]Pacific Northwest National Laboratory, Richland, WA, USA
[2]Helmholtz Centre for Environmental Research, Leipzig, Germany

*Correspondence to*: Jim Yoon (jim.yoon@pnnl.gov)

**Abstract.** Large-scale hydrological models (LHMs) are commonly used for regional and global assessment of future water shortage outcomes under climate and socioeconomic scenarios. The irrigation of croplands, which accounts for the lion's share of human water consumption, is critical in understanding these water shortage trajectories. Despite irrigation's defining role, LHM frameworks typically impose trajectories of land use that underlie irrigation demand, neglecting potential dynamic feedbacks in the form of human instigation of and subsequent adaptation to water shortage via irrigated crop area changes. We extend an LHM, MOSART-WM, with adaptive farmer agents, applying the model to the Continental United States to explore water shortage outcomes that emerge from the interplay between hydrologic-driven surface water availability, reservoir management, and farmer irrigated crop area adaptation. The extended modeling framework is used to conduct a hypothetical computational experiment comparing differences between a model run with and without the incorporation of adaptive farmer agents. These comparative simulations reveal that accounting for farmer adaptation via irrigated crop area changes substantially alters modeled water shortage outcomes, with U.S.-wide annual water shortage reduced by as much as 42 percent when comparing adaptive and non-adaptive versions of the model forced with U.S. climatology from 1950-2009.

## 1 Introduction

Threats to water security are a paramount global concern, driven by growing demographic pressures on scarce water resources and a changing climate (Vorosmarty et al., 2000; Oki and Kanae; 2006; Vorosmarty et al., 2010; Schewe et al., 2014; Liu et al., 2017; Huang et al., 2019). Regional and global water security outcomes are commonly framed in terms of depletion to groundwater (GW) and surface water (SW) resources (Wada et al., 2010) or through water shortage, defined as the gap between water resources availability and human water demand (Hoekstra et al., 2012; Brauman et al., 2016). In modeling studies evaluating water shortage, human water demand is commonly identified as the primary driver in undesirable future water security outcomes (Vorosmarty et al., 2000; Hejazi et al. 2015; Voisin et al. 2016; Hadjimichael et al., 2020; Yoon et al., 2021).

Of the activities underlying human water consumption, irrigation accounts for the lion's share (Doll and Siebert, 2002; Brauman et al., 2016, Huang et al., 2018). Over the 20th century, the pace of irrigation expansion has been remarkable, with a six-fold increase in irrigated areas (Siebert et al., 2015). Modelling studies estimate that ~$2,700 \pm 540$ km$^3$ of water is withdrawn globally each year for irrigation and ~$1,200 \pm 99$ km$^3$ of that water consumed (McDermid et al., 2023), though such estimates of global irrigation are prone to considerable uncertainty (Puy et al., 2022). Based on a country-specific

estimate for the United States (Dieter et al,, 2018), irrigation water withdrawals were estimated at 163 km$^3$ in 2015, with consumptive use for irrigation estimated at 101 km$^3$. While climate change threatens the availability of water supply to sustain current irrigation practices (Elliot et al., 2014) and inter-sectoral competition for water resources may also limit irrigation potential (Rosegrant et al., 2002), opportunities for sustainable irrigation expansion have also been identified to enhance food security under both current and future climatic conditions (Rosa et al., 2018; Rosa et al., 2020a; Rosa et al.,

2020b; Rosa, 2022). Dams have been noted to play a unique and critical role in realizing this future irrigation potential (Schmitt et al., 2022). The role of dams on the enablement of irrigation has been a particular focus of several large-scale modeling analyses (Hanasaki et al., 2006; Fekete et al., 2010; Biemans et al., 2011; Voisin et al., 2013a; Haddeland et al., 2014).

Despite irrigation's defining role, existing large-scale hydrological modeling (LHM) frameworks for national to global assessment of water shortage (Vorosmarty et al., 2000; Doll et al., 2003; Hanasaki et al., 2008; Pokhrel et al., 2016; Sutanudjaja et al., 2018; Grogan et al., 2022) often exogenously impose trajectories of human land use that underlie irrigation demand in models, neglecting potential natural to human system feedbacks (Wada et al., 2017; Huang et al., 2019) such as human instigation of and subsequent adaptation to water scarcity (Turner et al., 2019; Dolan et al., 2021). Regional

analyses for example indicate that drought conditions and reduced water supplies can lead to fallowing of land and reduction of irrigated areas, such as during the 2012-2016 California drought (Howitt et al., 2014). Evaluation of the interplay between surface water reservoir management and irrigation demand via the "reservoir effect" (Baldassarre et al., 2018) and similar dynamics is also precluded in large-scale water shortage analysis due to the lack of dynamic cropping adaptation in LHMs. While cropping "migrations" have recently garnered attention in the agricultural climate adaptation literature (Sloat et al.,

2020), the inability of LHMs to represent such dynamics risks potential misdiagnosis of water shortage outcomes. The representation of dynamic irrigated crop area adaptation in LHM frameworks for national to global scale water shortage analysis remains a major gap, though recent regional agent-based LHM implementations suggest the potential (De Bruijn, J. A. et al., 2023)

Our effort builds upon local and regional water modelling studies that have introduced and developed various forms of two-way coupling between agent-based models and hydrologic water systems models over recent years (Reeves and Zellner, 2010; Giuliani et al., 2016; Castilla-Rho et al., 2017; Khan et al., 2017; Yang et al., 2018; Hyun et al., 2019; Yang et al, 2019; Yoon et al., 2021; Lin et al., 2022; Klassert et al., 2023). Most of these local and regional applications have focused on

capturing coupled human-hydrological interactions in an irrigated agricultural context, with some also including representation of dam operation and other water user agent categories (e.g., urban user agents). In these previous efforts, agent-based models have been integrated with models that are commonly used for case-specific representation of hydrological water systems, such as agent-based model integrations with MODFLOW (Reeves and Zellner, 2010; Yoon et al., 2021), SWAT (Khan et al., 2017), and Riverware (Hyun et al., 2019). The coupling of an agent-based model with a *large-scale* hydrological model distinguishes the current effort.

Here we present a new modeling framework, WM-ABM, to evaluate water shortage outcomes that emerge from the interplay between climate-driven surface water availability, reservoir management, and farmer irrigated crop area adaptation. The integrated model extends a large-scale grid-based river routing, reservoir management, and water allocation model, MOSART-WM (Voisin et al, 2013b), with a multi-agent farmer model of crop choice developed using a positive mathematical programming (PMP) approach (Howitt, 1995). While the PMP is an optimization-based simulation model, the approach allows for automated calibration to observed cropping, economic, and hydrologic data, capturing realistic crop patterning of farms (Heckelei et al., 2012) and flexibly accommodating local, regional, or national calibration datasets. The model is deployed to conduct a hypothetical computational experiment at 1/8th degree (~12 km) spatial resolution over the continental United States (CONUS) (~50,000 grid cells),

## 2 Methods

### 2.1 Integrated Modeling Approach

The large-scale spatially distributed modeling framework, WM-ABM, integrates a farmer cropping adaptation model into a large-scale river routing water management model. For the latter, an agent-based model (ABM) approach is adopted with a representative farmer implemented for each model grid cell. The farmer agent crop selection and irrigation decisions are based on a Positive Mathematical Programming (PMP) approach, a method for calibrating agricultural production functions to observed data. Farmer decisions are based on water availability provided by the large-scale water management model, MOSART-WM, which simulates surface water availability for irrigation. The ABM and MOSART-WM models exchange information on an annual basis, with farmers looking to past simulated water availability from MOSART-WM at the onset of each calendar year and providing an updated water demand based on cropping decisions to MOSART-WM for the upcoming year. Following this annual update of demand, MOSART-WM proceeds on a daily timestep until the following calendar year, at which time farmers once again update their water availability forecasts, crop areas, and irrigation demand. A schematic illustrating a single year of the model run is shown on Figure 1. The framework is applied to the continental United States at 1/8th degree spatial and daily temporal resolution, with model output aggregated and reported on an annual and monthly basis. The two primary sub-models of the integrated model, the farmer cropping sub-model and the water availability sub-model, along with their coupling and various data inputs are described further in the following sections.

## 2.2 Agent-based Model of Farmer Cropping Decisions

The ABM entails agricultural agents, where each agent serves as an aggregated farm, representing all real-world farms that are located within a 1/8 degree grid cell (resulting in 53,835 representative farm agents over the entire model domain). The agents are involved in determining the types of crops to be grown over the 1/8 degree grid cell and the areas for each crop,

taking into consideration the associated water requirements to meet the irrigation needs for the selected crop patterning. The agents further determine how much surface water and groundwater to use for irrigation based on the relative cost and availability of each water source. As the MOSART-WM model focuses on simulating surface water availability, groundwater availability for irrigation is assumed to remain steady at the availability and cost estimated for the baseline period. For example, under an increase in surface water availability farm agents can respond by either reducing their

groundwater production, or increasing their irrigated cropped areas while maintaining the same level of groundwater production. While groundwater is treated as in infinite reservoir at a static groundwater level over the simulation period, groundwater production for any given annual time step is constrained to the amount of groundwater production estimated for the calibration period. Our representation of groundwater in the model and its limitations are further addressed in the Discussion section.


Farmers update their crop choices on an annual basis at the start of the calendar year based on an imperfect forecast of future surface water conditions for the coming year. For the current experiment, farms forecast future surface water availability for irrigation through tracking the state of hydrological proxies from MOSART-WM, which are then processed through a memory decay function (Tamburino et al., 2020) to determine a forecasted water demand for the following year. This annual

demand forecast is further partitioned to individual months following the water use disaggregation method and dataset described in Moore et al. (2015), which utilizes a phenological approach to disaggregate annual irrigation water demand to monthly demand at 1/8 degree resolution over CONUS. Specifically, farmers adopt the following steps at the start of each model year (January 1):

1. Identify the average state of the hydrological proxy for the most recent simulated year ($H_t$).
    a. For farmers within 4 grid cells (~50 km) of a river impounded by reservoirs (see MOSART-WM section below), the hydrological proxy is the sum (and associated annual deviations) of simulated storage volumes across upstream reservoirs impounding this river. The 50 km threshold represents a reasonable estimate of a distance cutoff for most diversions (Biemans et al., 2011), while also aimed at reducing computational

expense (as the buffer increases, additional agents/cells have access to any given reservoir, increasing the computational requirement for the reservoir water allocation algorithm).
    b. For farmers not relying on upstream reservoirs, the hydrological proxy is the river discharge in the coincident MOSART-WM grid cell.

2.  Determine an adjusted surface water demand (Dem$_{adj}$) by dividing the updated hydrological proxy by the hydrological proxy during the calibration period (H$_b$) and multiplying by the surface water demand during the calibration period (Dem$_b$). The hydrological proxy during the calibration period can be viewed as a long-term average of the hydrological state identified in step 1 above.

$$Dem_{adj} = (H_t/H_b) * Dem_b$$

3.  Process the adjusted surface water demand through a memory decay equation to calculate forecasted surface water demand (Dem$_f$)

$$Dem_f = \left[(1 - \mu) \times Dem_{f,t-1}\right] + (\mu \times Dem_{adj})$$

**where**:

μ - memory decay factor between 0 and 1

Dem$_{f,t-1}$ – surface water demand forecast from previous timestep (or Dem$_b$ if initial timestep)

The memory decay factor, μ, determines how much the farmer agent weighs distant versus recent experience (with higher values indicating a higher weighting of recent experience). As a default, the factor is set at 0.2 (which weights the most recent year by 0.20, the year before that by 0.18, and so forth) with different values explored during sensitivity analysis. A figure illustrating the effect of μ on the relative influence of previous year experience is included in the supplemental materials.

For the implementation of adaptive crop choice and irrigation decision making under dynamic expectations of water availability, agricultural agents are assumed to behave as profit maximizing firms, implemented and calibrated using a positive mathematical programming (PMP) approach. The PMP approach, introduced by Howitt (1995) has been widely using in agricultural policy modeling studies (de Frahan et al., 2007; Heckelei et al., 2012). In the adopted PMP framework, an agricultural agent's crop choice decisions, including types of crops and areas (and associated crop production inputs), are framed as a quadratic optimization problem in which the farm agent maximizes profit subject to land and water availability constraints.

Two PMP calibration coefficients (one linear and one quadratic) are added to the profit maximization formulation. The coefficients account for the increasing marginal cost of expanding the production of any crop in a given region, due to limited local availability of crop-specific labor and capital endowments (e.g., specialized machinery, skilled workers, and farmers' knowledge), as well as heterogeneous environmental, land, and marketing conditions (Heckelei et al., 2012; Howitt et al., 2012). These increasing costs cannot be derived directly from available data on the average regional production costs,

but are revealed in farmers' crop allocation decisions (Paris, 2012; Paris and Howitt, 1998). The PMP approach utilizes this by calibrating the "unobserved cost" coefficients to observed historical crop acreage data (Garnache et al., 2017).

The PMP procedure generally follows a two-phase process. In the first calibration phase, the coefficients for the unobserved cost terms are solved using a linear optimization such that observed crop areas are reproduced under known historical conditions (i.e., known conditions of production costs, prices, land availability, and water availability). The development of input data sets for this calibration phase are described further in the following sub-sections. In the second simulation phase, the calibrated optimization model is then used to simulate farm agent cropping decisions for scenarios which include

changes in economic (e.g., costs and prices) and physical (e.g., water availability) conditions, in our case the simulation phase entailing the coupled MOSART-WM runs. Specifically, each agricultural agent seeks to maximize profit according to the following formulation:


$$
\begin{aligned}
\textbf{maximize}: profit = \sum_i (price_i \times yield_i \times areairrtotal_i) - (landcost_i \times areairrtotal_i) - \\
(\alpha_i \times areairrtotal_i) - (0.5 \times \beta_i \times areairrtotal_i^2) - \\
(swcost \times areairrsw_i) - (gwcost \times areairrgw_i)
\end{aligned}
$$

$$
\begin{aligned}
\textbf{subject to}: \sum_i areairrtotal_i &\leq availablearea \\
cir_i \times areairrsw_i &\leq swavailability \\
cir_i \times areairrgw_i &\leq gwavailability \\
areairrsw_i + areairrgw_i &= areairrtotal_i
\end{aligned}
$$


**where**:

$i$ – index for crop categories

price – unit farm-gate price which farmers receive for the sale of crop i ($/ton)

yield – amount of crop produced per land area of crop planted (ton/acre)

areairrtotal – total irrigated planted crop area (acres)

areairrgw – total planted crop area irrigated with groundwater (acres)

areairrsw – total planted crop area irrigated with surface water (acres)

cir – crop irrigation requirement (cubic meters / acre)

landcost – unit cost for land-based inputs excluding water ($/acre)

swcost – unit cost for surface water ($/acre)

gwcost – unit cost for groundwater ($/acre) (fixed at the groundwater cost estimated for the calibration period).

$\alpha$ – first PMP calibration coefficient

$\beta$ – second PMP calibration coefficient

swavailability – expected surface water availability (as processed through MOSART-WM)

gwavailability – expected groundwater availability (set at groundwater production volume estimated for the calibration period)

The first term in the profit maximization formulation above represents the farmer's revenues from crop sales, the second term represents known land-based costs for producing crops, the third and fourth terms are the calibrated unobserved crop production costs, and the fifth and sixth terms are costs for surface water and groundwater production for irrigation. The unobserved costs have a modulating effect on the degree of simulated crop divergence from historical patterns, tending to "pull" agents towards selecting crops observed in the calibration period while changes in modeled conditions (prices, resource availability, etc.) potentially "push" agents towards new crops. The data sources and processing for the agent calibration are described further in the "Farm Data for PMP Calibration" sub-section below.

## 2.3 Farm Data for PMP Calibration

The first stage of the PMP model development process involves calibration of agricultural agents' unobserved cost coefficients over a historical period. Specifically, coefficients for the unobserved cost terms in the farm agents' profit maximization formulation are calibrated such that the agents reproduce observed crop areas under a known set of historical prices, costs, and resource constraints (e.g., water and land availability). For the current effort, we aggregate farm agents at 1/8th degree grid resolution over the continental United States, following the North American Land Data Assimilation System (NLDAS) grid. In view of future work evaluating global change conditions, we adopt crop categories from the Global Change Analysis Model (GCAM) (Calvin et al. 2019), a country-scale and global scale multisectoral partial equilibrium model used by the community to answer what-if science questions around energy-water-land interactions under policy, climate, and technology change.

The PMP calibration procedure assumes that observed crop areas are the outcome of conditions representative of a specific period of time. As such, the PMP calibration benefits from identifying data sources that are available for a coincident time period. Given the large spatial extent of our model, data for the PMP calibration has been drawn from several disparate data sources that often do not precisely align in terms of the date of data collection. The data sources used for PMP calibration are summarized in Table 1. For our purposes, we assume that these various data sources are an averaged representation of the 2010-2013 historical period, though we recognize that the data sets are drawn from different years and that conditions may be variable within this time period. We select the 2010-2013 period due to both data availability as well as historic drought conditions: 1) the Cropland Data Layer (CDL), a critical input for a our data workflow, is only available starting in 2008, 2) the start of the 2010s were a period in which historic drought over the United States was relatively low as a baseline (i.e., we are not calibrating to an atypical period of water shortage), and 3) the USDA Farm and Ranch Irrigation Survey is only available in 2013. We combine these data sources together (CDL starting in 2010) and consider them a historic representation of 2010-2013 conditions. Exploring the sensitivity of PMP parameters and model behaviour to the choice and uncertainty of these input datasets is an important future research direction. For example, the PMP could be recalibrated

based on different years in which the CDL and USDA Farm and Ranch Irrigation Survey is available, assessing the sensitivity of the calibrated PMP coefficients against those generated with the data used for this particular study.

To calibrate the PMP model, estimates of the following data at 1/8 degree resolution for each GCAM crop category representative of the historical calibration period (2010-2013) are required: crop land area, crop prices, crop land-based

costs, water-based costs, crop yield, and proportion of cropped area that is irrigated. Development of these data sets based upon United States Department of Agriculture (USDA) and other agricultural data sources is described further in the sub-sections to follow. We further note that we largely rely on nationally-available datasets for consistency in calibration input across the model domain, though the PMP automated calibration approach could readily accommodate other datasets (e.g., local agricultural datasets).

### 2.3.1 Crop Land Area Data for PMP Calibration


Observed cropped land area data for the continental United States used during the PMP calibration are estimated at 1/8 degree resolution combining data from the 2013 USDA Farm and Ranch Irrigation Survey (FRIS) (USDA, 2013), which reports official cropped areas at the state-level, and the Cropland Data Layer (CDL) (USDA, 2019), which provides additional estimates of observed land cover classifications at 30-meter resolution across the continental United States on an

annual basis since 2008. To further determine proportions of irrigated versus non-irrigated crops and for those crops that are irrigated, proportions of groundwater irrigated to surface water irrigated crops, we leverage the FAO's global map of irrigated areas (Siebert et al., 2013). The approach generally relies on the USDA irrigation survey data for acres irrigated by crop at the state level, then distributes these total areas within a state at 1/8 degree resolution following the distribution observed in the CDL data. The FAO global map of irrigated areas is used to further distinguish cropped areas by the portion

of area that is non-irrigated versus irrigated, and for the latter, the portion of area that is irrigated with surface water versus groundwater.

As the FRIS, CDL, and GCAM crop categories differ, the first step in data processing involves developing a mapping between the crop categories for each source (included in the supplemental materials). Using the mapping, each FRIS and

CDL crop category is assigned to a more general GCAM crop category. Subsequently, the area of irrigated crops (following the GCAM crop categories), distinguished by surface water and groundwater irrigation, are calculated for each 1/8th degree grid cell using the following general sequence (for each State):

1. Determine summed area of CDL pixels assigned to each crop category for each 1/8 degree grid cell.


2. For each grid cell, determine the area of each crop category that is irrigated with groundwater, irrigated with surface-water, and non-irrigated based on percentages reported in the FAO global map of irrigated areas dataset.

3. Sum the total irrigated area for all crop categories across the State.

4.  Apply a scaling correction factor (uniform for all 1/8 degree grid cells across the State) to the irrigated areas calculated in step 2, such that the total State-wide irrigated area (calculated in step 3) matches the irrigated areas reported in the USDA Farm Ranch and Irrigation Survey.

The full implementation and code is provided in the project meta-repository. The approach assumes that the USDA 2013 FRIS dataset reported at the state-level is an accurate representation of total area of cropped land across the state, while the CDL maps which are based upon classification of satellite images effectively capture the spatial distribution of cropped area within a state (though are a less reliable indicator of total cropped area compared to the USDA irrigation survey) and the FAO global irrigation maps provide a reasonable spatial distribution of irrigated vs. non-irrigated and groundwater vs. surface water irrigated areas.

In some cases, the method above results in total crop areas that exceed the total available land area of a 1/8 degree grid cell. For these cells, we set the available land area constraints in the agent-level optimization (see above) to this total observed crop area such that the PMP can reproduce the calculated total land areas but are unable to exceed them in simulation mode.

For land areas constraints set for each farm, we assume that areas assigned to the following land use categories based upon the CDL remain fixed throughout the model run and are unavailable for cropping: "NotAvailable", "RockIceDesert", and "UrbanLand". To determine the land area constraint that enters the PMP formulation for each NLDAS grid cell, we take the maximum of: 1) The total land area in the grid cell subtracted by the categories described above and 2) the total land area allocated to irrigated crops determined from the data processing workflow described above.

## 2.3.2 Baseline Water Demand and Irrigation Estimation

The water demand estimation for the baseline calibration period is calculated by taking observed crop-specific irrigated areas (see section above) and multiplying these by a region-specific irrigation requirement based on the 2013 USDA Farm and Ranch Irrigation Survey (FRIS) (USDA, 2013). Specifically, the average acre-ft/year of irrigation water applied per acre of land is obtained for each state on a crop-specific basis, which is assumed to be consistent with the crop irrigation requirement. For each 1/8 degree grid cell, the baseline demand is subsequently determined by multiplying the estimated irrigated crop areas with their associated crop irrigation requirements and summing across all crops. A table of the state-level irrigation requirements on a crop-specific basis is included in the project meta-repository. In an adaptive model simulation, agents are initialized with their baseline water demands, and subsequently adjust their water demands as they adapt to changing water availability conditions as the model steps through time. In our formulation, only cropped areas are assumed to change over time (the irrigation requirement, i.e., the depth of irrigation water required per unit land area of crop planted is assumed to remain static over the model run). As the baseline water demands are based on actual applied water data, these demands are assumed to account for state-to-state and crop-to-crop variation in climatology, irrigation technology, water use efficiency, and other factors that influence crop irrigation requirements. The baseline water demand and irrigation estimation

is segmented by surface water and groundwater based on the source-specific estimated irrigated areas estimations described in Section 2.3.1. For groundwater specifically, this baseline groundwater irrigation estimate provides an agent-specific annual groundwater production cap (in acre-ft/year) that the agent cannot exceed for any given year of the model run.

### 2.3.3 Crop Prices, Costs, and Yields

Crop prices and costs are obtained from the USDA Economic Research Service's (ERS) commodity costs and returns datasets, which are produced by the USDA on an annual basis. Prices and costs in these datasets are aggregated to 9 ERS farm resource regions across the United States. Similar to the USDA irrigation survey, ERS crop categories are mapped to more general crop categories to enhance compatibility with other global models (as detailed in Table S1 in the supplemental materials). Each NLDAS grid cell then uses the economic information of ERS farm region that it is located within. Specific economic price and cost data derived from the ERS datasets for each ERS farm region and GCAM crop category include: total cost of production ($ / acre), crop yield (bushels or tons / acre), the opportunity cost of labor ($ / acre), and the opportunity cost of land ($ / acre). For the acreage-based inputs, the ERS costs are generally estimated through surveys that ask the farmer how much was paid on a per acre basis for the various inputs, which we assume to be on a planted area basis. For empirical reasons we remove the USDA estimates of unpaid labor costs and imputed opportunity costs of land, as the PMP calibration terms are better able to capture the relevance and heterogeneity of these non-monetary production costs. Crop prices and production costs are static over the simulation period. The full cost tables are included as part of the data and code meta-repository included with this manuscript.

### 2.3.4 Irrigation Water Sources and Costs

Irrigation water sources and costs are derived from the 2013 FRIS database. The survey provides state-level estimates of irrigation water volumes assigned to three water source categories: 1) groundwater, 2) on-farm surface water and, 3) off-farm surface water. The FRIS database also provides state-level estimates of water purchase costs for off-farm surface water ($ / acre-ft) and average energy pumping costs for farms that utilize on-farm groundwater for irrigation ($ / acre-ft). These per unit water production costs are assigned to each NLDAS grid cell based upon the state that the cell falls within. On-farm surface water is assumed to be free. Irrigation water costs are static over the simulation period.

### 2.3.5 Partitioning Land and Water Based Costs

To partition land and water based costs in our farm agent's economic profit formulation, we adopt the following procedure reconciling data reported in the ERS commodity costs and returns dataset and the 2013 FRIS database (the latter which reports costs specifically for irrigation water supply), while also enforcing a minimum agent profitability threshold:

1. Calculate perceived crop production costs as:

$$PerceivedCost_{n,c} = TotalCost_{n,c} - OppCostLabor_{n,c} - OppCostLand_{n,c}$$

where:

PerceivedCost – The perceived cost of production ($/acre)

TotalCost – The total cost of crop production as reported in 2013 FRIS ($/acre)

OppCostLabor – The opportunity cost of labor for crop production as reported in 2013 FRIS ($/acre)

OppCostLand – The opportunity cost of land for crop production as reported in 2013 FRIS ($/acre)

2. Calculate the perceived profit for crops:

$$Profit_{n,c} = (Yield_{n,c} * Price_{n,c}) - PerceivedCost_{n,c}$$

where:

Profit – The profit obtained from selling crops ($/acre)

Yield – The yield of crop production as reported in 2013 FRIS (ton/acre)

Price – The farm gate price for selling crops as reported in 2013 FRIS ($/ton)

3. Calculate an adjusted perceived cost of production (PerceivedCostAdj) which forces a minimum profit margin of 10 percent. This step is applied such that all observed crop production is assumed to be profitable.

$$PerceivedCostAdj_{n,c} = (Yield_{n,c} * Price_{n,c}) - Profit_{n,c} \quad when \quad Profit_{n,c} \geq .10 * Yield_{n,c} * Price_{n,c}$$

$$PerceivedCostAdj_{n,c} = (.90 * Yield_{n,c} * Price_{n,c}) \quad when \quad Profit_{n,c} < .10 * Yield_{n,c} * Price_{n,c}$$

4. Partition total crop production costs into land based costs and water based costs assuming:

$$PerceivedCostAdj = LandCost + GroundwaterCost + SurfaceWaterCost$$

where:

LandCost – Land based crop production costs ($/acre)

GroundwaterCost – Cost of groundwater based upon 2013 FRIS ($/acre)

SurfaceWaterCost– Cost of surface water based upon 2013 FRIS ($/acre)

5. For instances in which the sum of groundwater cost (GroundwaterCost) and surface water cost (SurfaceWaterCost) exceeds the total cost of crop production as reported in 2013 FRIS, adjust the GroundwaterCost and SurfaceWaterCost to assume that they comprise the same proportion of the total cost of crop production as the United States average.

**Large-Scale Modeling of Water Availability**

Surface water availability that feeds into the farm agents' cropping and irrigation decisions is dynamically provided by MOSART-WM. MOSART-WM is a spatially distributed large-scale water management model consisting of a physically based river-routing model (MOSART, Li et al., 2013) coupled with a generalized water-management model (WM, Voisin et al. 2013b) for seasonal to long term studies. MOSART-WM takes surface runoff generation input from an external hydrological model, commonly the Variable Infiltration Capacity (VIC) model in previous applications. In the river-routing

component, daily surface runoff is an input that is first routed across hill slopes and then discharged into a tributary subnetwork within each grid cell before entering the main channel for transport across grid cells. WM has two components – reservoir operations which influence the seasonality of flow and river storage, and water supply management which allocates supply from reservoirs to spatially distributed demands across grid cells.

Our CONUS set up of MOSART-WM model includes all (1,848) reservoirs with a storage capacity larger than 10 million m$^3$, i.e. focusing on reservoirs that most influence river discharge. The reservoir database and locations are obtained from the Global Reservoir and Dam Database (GranD) reservoir database (Lehner et al., 2011). For daily reservoir storage and release operations, MOSART-WM adopts generic operating rules that mimic monthly release and storage patterns based on the objective of the reservoir (e.g., flood control, irrigation, etc.), its physical characteristics (storage) and monthly climatologies of inflow and demand, and follows daily constraints for minimum environmental flow, and minimum and maximum storage

volumes. The reservoir model builds upon generic operating rules introduced in Biemans et al. (2011) and Hanasaki et al. (2006) and improved upon for multi-objective operations by Voisin et al. (2013b).

During the calibration phase of the PMP model, we assume that agent's water availability constraints are non-binding (i.e.,

the irrigation water required to produce the observed surface-water irrigated crop area estimates were available during the calibration period). To account for potential inconsistencies between the estimated surface water irrigation demand (as estimated via the data sources and procedures described above) and VIC-MOSART-WM simulated irrigation water availability for the calibration period, we then apply a bias correction factor to the crop irrigation requirements on a cell-by-cell basis such that total irrigation demand matches VIC-MOSART-WM simulated water availability for the historical

calibration period. Such a treatment attempts to reconcile potential inconsistencies between estimated irrigation requirement calculated from the data assimilation process described throughout the sub-sections above and actual irrigation water availability modeled by VIC-MOSART-WM for the baseline period.  The additive cell-specific bias correction factor is subsequently applied in simulation mode for each time period. This approach addresses potential biases in VIC-MOSART-WM's estimates of irrigation water availability for baseline conditions, and assumes these biases are maintained in the ABM

simulations that depart from baseline conditions while relying on modeled results to estimate changes in water availability relative to the baseline condition.

## 3 Design of Computational Experiment

To evaluate the impact of farming cropping adaptation on model outcomes, the model is deployed to conduct comparative simulations at 1/8th degree spatial resolution across CONUS with and without incorporation of the adaptive farmer agents.

To capture a realistic sequence of hydrologic conditions, we conduct an experiment that mimics the 1950-2009 hydrological record, using simulated runoff derived with the VIC hydrology model (Liang et al., 1994). Henceforth, we number model years between 1-60, referring to a model year as MY and its associated historical hydrological calendar year as CY (e.g., MY 1 is associated with CY 1950). The CY 1950-2009 period is specifically selected to capture a range of hydrologic conditions as VIC outputs are available over this time period for a simulation that has been calibrated and is considered a benchmark for

the United States Bureau of Reclamation (USBR), as used in the Secure Water Act (see Reclamation, 2014 for additional details on VIC simulations and calibration results).

To distinguish the effect of farmer adaptation to hydrological change on water shortage outcomes, we compare model results between a non-adaptive run and an adaptive run, herein referred to as the "baseline" and "adaptive" runs respectively. Water

shortage is calculated as:

$$WaterShortage = WaterDemand - LocalWaterSupply - ReservoirWaterSupply$$

where:

WaterDemand – The initial water demand for irrigation water as determined by a farmer agent

LocalWaterSupply – Irrigation water supply to an agent from river flow in a coincident grid cell (via MOSART-WM)

ReservoirWaterSupply – Irrigation water supply to an agent from connected reservoir(s) (via MOSART-WM)

In the baseline run, the surface water and groundwater irrigated crop areas are exogenous and static (to the baseline 2010-

2013 conditions), as is common in the LHM literature. In the adaptive run, farmer agents are initialized using the baseline 2010-2013 conditions and subsequently endogenously determine their irrigated cropped areas for 10 general crop categories on an annual basis based on dynamically updated expectations of surface water availability (both local runoff and reservoir storage). The water shortage analytics for both runs are then calculated using unmet demand for surface water (i.e., the amount of water demanded subtracted by the amount of water supplied). For the adaptive run, expected agricultural profits

are also calculated for each year based on the farms' cropping decisions. With the adaptive farmer agents online, we additionally conduct sensitivity runs where we adjust the farmer agent memory decay parameter allowing us to evaluate the impact of the strength of agent memory on water shortage outcomes.

We note that the study is designed as a hypothetical experiment to evaluate the influence of farmer cropping adaptation on

modeled water shortage outcomes rather than an attempt at a historical reconstruction of actual cropping and water use

patterns, as other non-hydrological influences such as crop prices, areas equipped with irrigation, and crop-specific irrigation requirements remain static over the model run. The hypothetical experiment is rather designed to identify the potential cropping adaptation response to hydrologically-driven changes in water availability (using the 1950-2009 record as a reasonable window of hydrological variability), holding all other influences constant. As such, a detailed comparison of model results against observed data is not applicable, though we evaluate the plausibility and reasonability of model results by stress testing the farmer agent model as well as comparing modeled land use changes with observed land use changes over an isolated period of drought in the Western United States to determine whether modeled crop adaptations are commensurate with historical observations (see supplemental materials).

## 4 Results

### 4.1 The Influence of Farmer Cropping Adaptation on Water Shortage Outcomes

Accounting for farmer irrigated crop area adaptation substantially alleviates simulated annual water shortages, especially during periods of severe regional drought (Fig. 2, see Fig. S2 for monthly details). In Figure 2, we show the annual difference in water shortage between the adaptive and baseline versions of the model (annual water shortage in the adaptive run subtracted by annual water shortage in the baseline run), aggregated for four HUC 2 regions. We also identify the peak annual water shortage (across all model years) for farm agents across the western United States for both the adaptive and baseline runs, with the ratio of peak annual water shortage of the adaptive and baseline runs (i.e., peak annual water shortage of the adaptive divided by that of the baseline) shown on the Fig. 2. Blue colors indicate reduced shortages with adaptation and orange colors indicate increased shortages with adaptation. Water shortage alleviation due to adaptation is especially prevalent across the Western United States (Fig. 2b) and most evident during periods of drought (Fig. 2a), with the California Hydrologic Unit Code (HUC) 2 region by far exhibiting the largest shortage differences due to adaptation. For example, water availability conditions from MY 38-43 (CY 1987-1992) were one of the driest in California's recorded climate history. From the onset of these dry conditions to their culmination, modeled results indicate that cropping adaptation increasingly reduces water shortages, reaching a decrease in average annual water shortage of ~107 $m^3$/s when comparing the adaptive run to the baseline for MY 42 (monthly shortage decrease reaches a peak of 299 $m^3$/s for August MY 1992). Neglecting farmer adaptation potentially overestimates water shortage by over a factor of 2 for MY 42-43 (the CY 1991-1992 drought years in California). Similar effects arise when accounting for farmer adaptation in response to the droughts in the Missouri region at the onset of MY 50 (CY 2000s), the Upper Colorado region in MY 27-28 (the CY 1976-1977), and the Pacific Northwest region MY 38-43 (CY 1987-1992 Snake River low flow period). In the Missouri HUC 2 region decreases in water shortage reach 2.4 $m^3$/s in MY 53 (CY 2002) when comparing the adaptive run to the baseline (or 32 percent of the shortage in the baseline run). In the Pacific Northwest region, water shortage is reduced by 15 $m^3$/s in MY 43 (or 66 percent of the shortage in the baseline run) due to the preceding 5 year low flow period (CY 1987-1992 low flow period in the Snake River basin).

The evaluation of water shortages on a monthly basis (see supplemental materials) indicates that water shortage differences
between the adaptive model runs are concentrated in the peak irrigation months in regions that regularly experience water
shortage. In the California region for example, water shortage difference between the adaptive and baseline runs in July
reaches a peak of 299 m$^3$/s for MY 43 (~3x the annual MY 43 shortage difference rate). For the Pacific Northwest region,
peak water shortage difference between the runs reaches 126 m$^3$/s in MY 43 (~8x the annual MY 43 shortage difference
rate).

While accounting for farmer irrigated crop area adaptation primarily results in lower water shortages across the Western
United States, there are also regions and periods in which cropping adaptation exacerbates simulated water shortage
compared to non-adaptation. Increases in simulated water shortage are notable in the headwaters of the Middle Gila River
Basin in Arizona, the Bear Watershed northeast of the Great Salt Lake in Utah, and the southern end of the Central Valley in
California. Such increases in peak shortages with adaptation are expected over regions which tend to experience sequences
of increasing water availability (along with increasing farmer expectations of water availability), interspersed by low water
availability years which result in more severe shortage due to farmers increased water availability expectations. We also note
that lack of representation of long-distance inter-basin transfers may account for these increases in peak shortage for regions
like the Southern Central Valley of California and the Middle Gila River basin in Arizona, a limitation which is addressed
further in the discussions.

Across the eastern United States (east of the Mississippi River), water shortage differences between the adaptive run and the
baseline run are subdued, with significant changes isolated to southern pockets of the Texas Gulf, Lower Mississippi, and
South-Atlantic Gulf regions (see Fig. S3 in the Supplemental Materials for a full CONUS map). For both runs, the U.S. wide
peak water shortage occurs in MY 28 (CY 1977), with a 364 m$^3$/s peak for the baseline run and a 332 m$^3$/s peak for the
adaptive run, a 32 m$^3$/s difference between the two.

## 4.2 Farmer Cropping Adaptation via Acreage Changes versus Crop Switching

In the most prominent regions of water shortage, the modeling experiment results indicate that farmers primarily adapt
through the contraction of irrigated cropped areas with crop switching playing a secondary role. Figure 3a-d shows simulated
changes in total SW-irrigated crop areas for four representative regions with prominent shortage (California, Missouri,
Upper Colorado, Pacific Northwest), while Figure 3e-h assigns farms to crop adaptation categories based on the amount of
crop adaptation simulated over the model period. For the latter, agents are assigned to one of four categories depending upon
the level of crop adaptation activity, conducted for each agent considering every year of model output: 1) "crop
expansion/contraction" if the ratio of an agent's annual minimum surface-water irrigated crop area is less than 80 percent of
the annual maximum surface-water irrigated crop area, 2) " crop switching" if the predominant crop's share of the total crop

makeup for any given agent (measured in terms of crop area) changes by at least 5 percent between any two years of the model run (which do not need to be consecutive), 3) "both" if the agent satisfies both criteria 1 and 2 above, and 4) "none" if the agent satisfies none of these criteria.

Model results indicate that irrigated crop adaptation activity is heavily concentrated in the Western United States, with relatively subdued activity in the Eastern United States outside of the Lower Mississippi and South Atlantic-Gulf regions (see Fig. S4 in the Supplemental Materials for a full CONUS map). Across the Western United States, irrigated crop adaptation largely takes the form of overall crop expansion and contraction, with notable hot spots of adaptation including the California Central Valley, Snake River Basin, and the Western Missouri regions.

In California, total SW-irrigated cropped areas fluctuate substantially. Preceding the MY 38-43 drought (CY 1987-1992), SW-irrigated crop areas reach a maximum of 3.6M acres in MY 36 dropping to 2.5M acres by the end of the drought in MY 42, a 31 percent decline in SW-irrigated areas over an 8-year period. Crop makeup (in terms of the percentage each crop accounts for among all SW-irrigated crops) remains steady, with a slight shift away from rice crops (a decrease from 13.7 percent of all SW-irrigated crops to 11.8 percent). Crop adaptation in response to shortage is pronounced in the Central
Valley (Fig. 3e), with farmer agents predominantly adapting to water shortage through crop area contractions (blue cells)

In the Missouri region, widespread declines in SW-irrigated areas are simulated over the course of the first 10 years of the model run (CY 1950s drought) across all crop categories. During this period, SW-irrigated areas peak at 3.2M acres in MY 4, decreasing 25 percent to 2.4M acres in MY 13. Over this period, the relative proportion of crops remains nearly stable, with slight decreases in fodder grass as a relative share of total SW-irrigated crops (dropping from 48 percent of SW-
510 irrigated crops in MY 4 to 43 percent in MY 13). The relative drop in fodder grass is accompanied by a relative increase in grains and miscellaneous crops. Additional major cycles of crop area contraction and expansion are observed over the simulation period (e.g., expansion starting around MY 45 and subsequent contraction starting around MY 50). The Upper Colorado similarly experiences a steady reduction of SW-irrigated areas over the first several years of the simulation, reaching a minimum during MY 27-28 (the CY 1976-1977 drought) followed by cycles of crop expansion and contraction
during the latter half of the simulation period.

For the Pacific Northwest region, marked contractions in SW-irrigated cropped areas as well as crop switching are simulated over the course of a major low-flow period in the Snake River basin from MY 38-43 (CY 1987-1992). From a SW-irrigated area of 4.3M acres in MY 38, SW-irrigated area drops to a minimum of 3.3M acres in MY 43, a 24 percent decline in area. Substantial crop switching is also simulated over this period, with fodder grass giving way to miscellaneous crop, root tuber,
and wheat in terms of relative share of total SW-irrigated crop areas.

Farmer adaptation to declining water availability via total SW-irrigated crop area contractions and crop switching lead to associated declines in expected agricultural profits, though these declines are typically more subdued than the associated

decrease in expected water availability or SW-irrigated crop areas. For the California region over the MY 36-44 (CY 1985-1993) period, expected agricultural profits (totaled over both SW and GW irrigated areas) decline from $1.84B dollars to $1.68B dollars (an 8 percent decline), in spite of a 31 percent decline in SW-irrigated areas. The subdued profit impact is attributable to both crop switching and groundwater availability remaining steady, which comprises a significant percentage of irrigation water needs in California. In the Missouri region over MY 4-13 (CY 1953-1962), agricultural profits decline from $2.53B in MY 4 to $2.50B in MY 13, a mere 1 percent decline in spite of a 25 percent decline in SW-irrigated crop areas, reflecting the predominant role of groundwater over surface water for irrigation water in this region. In the Pacific Northwest region, expected agricultural profits decrease from $1.60B in MY 38 (CY 1987) to $1.54B in MY 46 (CY 1995), a 5 percent decline associated with a 24 percent decline in SW-irrigated area. Results showing expected agricultural profits for all HUC 2 regions and model years are included in the Supplemental Materials (Figure S5).

**4.3 The sensitivity of water shortage outcomes to farmer agent's memory of water availability**

Figure 4 indicates the annual average shortage percentage (shortage divided by demand) for the baseline run (blue dots) and four sensitivity runs with varying agent memory parameterizations (red dots) for several HUC 2 watersheds. The model years are ranked in descending order by year of shortage (as simulated in the baseline run) to evaluate the impact of agent memory for varying relative levels of shortages. The four sensitivity runs serve to assess the robustness of results to varying strength of agent's memory to past water availability, with the adaptive run described in the results above ($\mu$02) colored in the darkest shade of red and indicating relatively "long" memory, while lighter shades of red indicate decreasing strength of agent memory (i.e., increased agent reactivity to more recent experiences of water availability with higher values of $\mu$).

Across regions, shortage in the baseline run typically falls on one extreme of the adaptive runs for a given year (i.e. for a given year and region, the four adaptivity runs are either consistently higher or consistently lower than the baseline run, with some exceptions). The spread in shortage between the adaptive runs for any given year and region is also typically well constrained, with the largest spread between the sensitivity runs occurring during the higher shortage years in the California, Upper Colorado, and Pacific Northwest regions. These findings indicate that the comparative evaluation between the baseline and adaptive runs (in terms of the direction of water shortage change between the two runs) is largely robust to assumptions regarding the strength of agent memory, though the parameterization of agent memory can control the magnitude of the shortage difference between the baseline and adaptive runs.

Comparing the baseline results versus the adaptive results, we further observe that the largest reductions in water shortage due to adaptation tend to occur in the years of highest of water shortage. California provides the most prominent example, in which the shortage difference between the baseline run and the adaptive runs is readily apparent during the 10 highest years of shortage (i.e., the first 10 years on the x-axis of Figure 4). Beyond the 15th highest shortage year in California, this pattern reverses with the adaptive runs typically indicating slightly higher shortages than the baseline run. Similar patterns are

observed for the Pacific Northwest and Upper Colorado regions. For the Missouri and South Atlantic-Gulf Regions, agent adaptation tends to lead to higher shortages, irrespective of the degree of shortage in any given year.

## 5 Discussion and Conclusion

We demonstrate the representation of farmer irrigated crop area adaptation to changing water availability in a large-scale hydrological modeling framework, with farmer agents' adaptivity specified using a positive economic calibration approach. The modeling framework allows for the evaluation of dynamic feedbacks between irrigated cropping areas, reservoir management, and water availability, and reveal that these interactions strongly shape national-scale water shortage outcomes. In a comparative hypothetical experiment conducted for all of CONUS, annual water shortages decrease by as much as 42 percent when accounting for farmer cropping adaptation, with differences due to adaptation even further pronounced in regions prone to water shortage such as California (where neglecting farmer adaptation results in an overestimate of water shortage by a factor of 2 during the year of highest shortage). Our hypothetical modeling experiment indicates that traditional large-scale modeling efforts that neglect adaptive cropping adaptation may be liable to misdiagnosis of water shortage outcomes. While farmer adaptation to decreasing surface water availability is accompanied by an associated loss in expected agricultural profit, this loss is buffered by the ability to switch crops and reallocate groundwater to more profitable crops to compensate for surface water shortages.

Particularly in agricultural hotspots with highly variable or declining water availability, the assumption of non-adaptive behavior most common in LHMs leads to overestimation of water shortages in our experiment. By isolating the effects of changing water availability on farmer cropping through comparing hypothetical adaptive and non-adaptive runs, our findings suggest that adaptation can significantly alleviate water shortage and put into question the severity of water shortage outcomes indicated by global water security analyses that neglect such adaptation, especially for regions such as the Western U.S. in which water availability is expected to decline due to climate change (Dettinger et al., 2015). For such regions with declining water availability trends, representative farm agents systematically overestimate water availability when they do not adapt their cropping based on changing hydrological conditions, resulting in higher shortages.

While initial sensitivity tests indicate that our results are robust to assumptions regarding agent memory of water availability, additional uncertainty characterization and sensitivity experiments on agent behavioral formulation and parameterization offer a promising direction for future research. Such uncertainty analysis could either be conducted at the level of the source PMP calibration data inputs, at the level of the PMP parameters themselves, or via alternate structural representations of farmer cropping decisions (Yoon et al., 2023). For example, estimation of the PMP parameters using alternative data sources or implementation in a data assimilation framework (Maneta and Howitt, 2014) could be conducted to further explore the implications of agent parameterization on water shortage outcomes.

While our study focuses on irrigated crop area changes, the modeling framework could further be extended to consider additional farm-level adaptive mechanisms and considerations such as deficit irrigation, adoption of technological innovations (e.g., crop varietals, improved irrigation technologies, etc.), and costs for crop switching. New typologies for representing such adaptive action in human systems models (Yoon et al, 2022) could serve to design and organize such formulations. The hydroeconomic formulation of farmer agent crop choices could further enable the investigation of economy-wide, multi-sector interactions. For example, future research could link WM-ABM with computable general equilibrium or partial equilibrium models, exploring economy-wide consequences and feedbacks of water shortage induced cropping adaptation (in the fashion of Turner et al., 2019; Dolan et al., 2021; Basheer, et al., 2021) that account for the hydrological, water management, and farmer processes represented in WM-ABM. Such feedbacks may serve to further alter water shortage induced crop area impacts (e.g., via crop price signals that shift crop production to more water rich areas).

We further note additional key limitations of the current modeling framework. While we represent groundwater production for irrigation in the modelling approach, the availability and cost of groundwater production is held fixed to baseline calibration conditions, a limitation of our analysis. The ability of farms to increase groundwater extraction in response to surface water shortage, as well as changes in the availability and cost of groundwater (e.g., due to depletion of groundwater in stressed aquifers) are not currently represented in our modeling framework. These potential responses may either mute the simulated water shortage changes (e.g., instances in which farmers increase groundwater pumping to accommodate surface water shortage) or heighten them (e.g., instances in which increasing groundwater depletion results in even higher shortages and ensuing adaptive responses). It is challenging to suggest a consistent implication of improving groundwater dynamics in the model; in some cases, farms may compensate surface water shortage by pumping more groundwater where this is physically and economically viable, while in other cases groundwater depletion may lead to reductions in production (due to changes in capacity, cost, or quality) that intertwine with surface water availability changes in a more complex manner. Evolution of such changes over time given regional hydrological and agronomic differences remains a major question. The incorporation of a dynamic sub-model for groundwater aquifer response to surface water conditions and groundwater pumping thus presents an important avenue for future research and model improvement, one that is anticipated as a future update to the initial version of the model introduced in this work.

The absence of inter-basin water transfers in the modeling framework further limits the analysis, with some instances of increased shortage with adaptation (e.g., in the southern California Central Valley and the Middle Gila River basin which are recipients of large inter-basin transfers in reality) appearing to be artifacts of this modeling limitation. In these regions, farms in the model base expectations of water availability on local runoff, while in reality they are able to receive water from distant water sources via inter-basin water transfers. We note that inter-basin transfers are largely absent in LHM applications (Wada et al., 2017) due to data limitations at large scales, and suggest that the development and incorporation of

national (e.g., Siddik et al., 2023) and global inter-basin transfer datasets offers a promising next step for continued LHM improvement. Recent efforts have further explored the effects of social norms on the influence of farmer behavior, for example evaluating the influence of social norms on farmer forecasts (Hu et al., 2006). Such considerations have been operationalized in coupled human-water system models, such as representing social norms on compliance in irrigated agricultural groundwater systems (Castilla-Rho et al., 2017) and beliefs about water supply conditions in a reservoir-

controlled river system (Lin et al., 2022). The incorporation of social norms effects into ABM integration with LHMs represents another future avenue of research, though considerations of aggregation and scale (e.g., highly aggregated representative agents in LHMs) pose additional conceptual challenges relative to more highly-resolved applications.

The coupled model and hypothetical experiment presented here intentionally focuses on isolating farmer cropping response

to surface water availability changes, tracing complex dynamic feedbacks between hydrological variability, surface water reservoir management, farmer crop choice, and irrigation demand. Our findings indicate that these interactions can shape the geographic distribution of crops and alter the magnitude of simulated water shortages, with the coupled MOSART-WM and farm ABM modeling framework providing a foundation from which to account for dynamic cropping adaptation amidst climate and socioeconomic change in future large-scale water security assessments.

**Author Contribution Statement**

JY: Conceptualization, Methodology, Investigation, Writing – original draft, Visualization. NV: Conceptualization, Methodology, Writing – review & editing. CK: Conceptualization, Methodology, Software, Writing – review & editing. TT – Software. WX – Software.

**Data and Code Availability Statement**

All data and code required to run the model and reproduce numerical experiments are provided in the following repository: https://github.com/IMMM-SFA/yoon-etal_2023_hess

**Acknowledgments**

This research was supported by the U.S. Department of Energy, Office of Science, as part of research in the MultiSector Dynamics, Earth and Environmental System Modeling Program (grant 59534). Pacific Northwest National Laboratory is a

multi-program national laboratory operated by Battelle for the U.S. Department of Energy under Contract DE-AC05-76RL01830. We thank Sean Turner and Jennie Rice for their valuable comments on an initial draft of the manuscript.

## Competing Interests

The authors declare no competing interests.

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

| Data/Survey Product Name | Data Description | Resolution | PMP Application |
|---|---|---|---|
| 2013 USDA Farm and Ranch Irrigation Survey | Irrigated crop areas | State-level | Used for state-level estimates of cropped areas which feed into downscaling process (see "Cropland Data Layer" entry below) |
| 2010 Cropland Data Layer | Observed land cover classifications across the continental United States | 30-m | The 30-meter land use classifications are used to downscale state-level irrigated crop areas to 1/8 degree resolution, which is input to the PMP calibration process |
| 2013 FAO Global Map of Irrigated Areas | Proportions of irrigated versus non-irrigated and groundwater versus surface water cropped areas | 5 arc minutes | Used to spatially distinguish between irrigated and non-irrigated crops. For irrigated crops, used to further distinguish between surface water and groundwater irrigation sources. |
| 2010 USDA Economic Research Service's (ERS) Commodity Costs and Returns | Estimates of annual production costs and returns based on historical accounts | 9 ERS Farm Resource Regions across the United States | Used to provide prices (e.g., crop prices) and costs (e.g., land-based costs) that are used during the PMP calibration process |
| 2013 USDA Farm and Ranch Irrigation Survey | Irrigation water costs separated by groundwater, on-farm surface water, off-farm surface water | State-level | Used to estimate water cost of different irrigation sources for input during the PMP calibration process |
| 2013 USDA Farm and Ranch Irrigation Survey | Irrigation water requirements per crop | State-level | Used to estimate irrigation water requirements per crop for input during the PMP calibration process |

**Table 1: Agricultural and irrigation data sources** for calibration of baseline PMP farm cropping model.

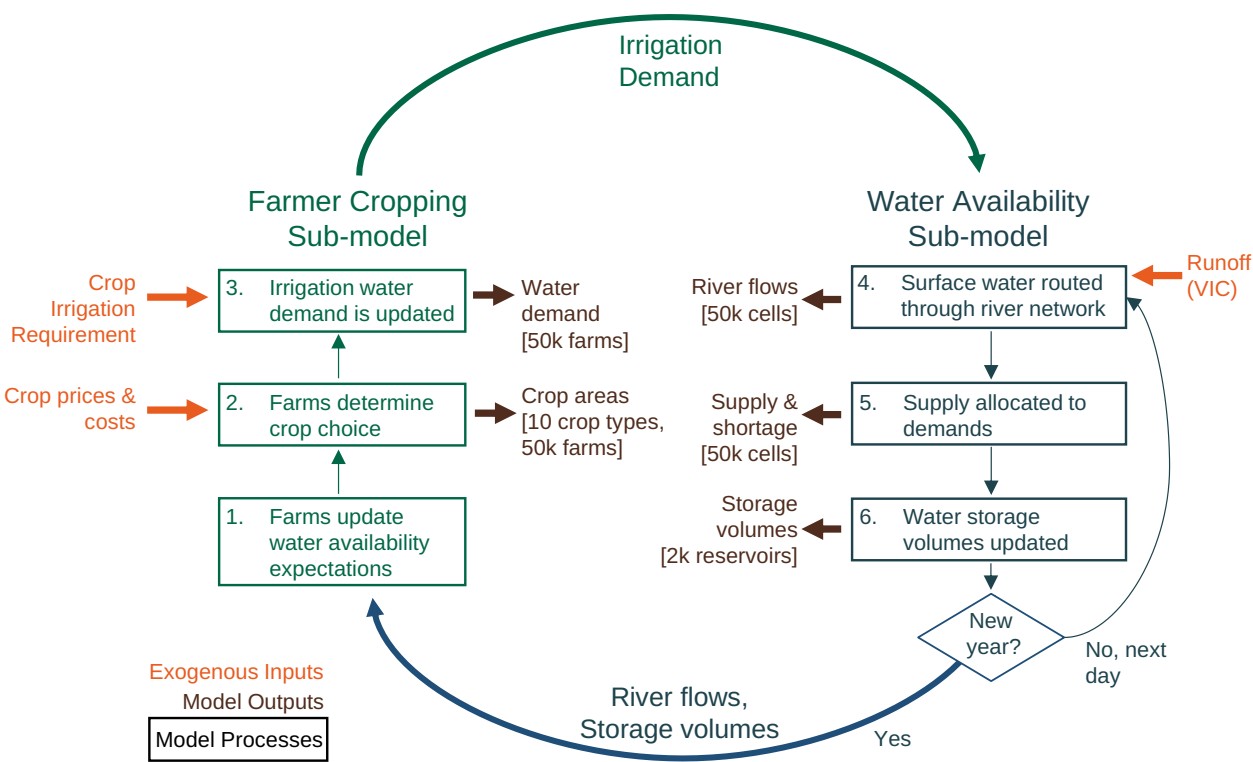

**Figure 1: A model schematic of WM-ABM** distinguishing between exogenous inputs (orange), model processes (green/blue), and model outputs (brown). The model processes are further distinguished between the two interacting sub-models that constitute the integrated model: the farmer cropping sub-model based on PMP and the water availability sub-model based on MOSART-WM. For any given year, farms first update water availability expectations (using dynamic flow and storage states provided from the water availability sub-model) and make crop choice decisions. These decisions result in updated irrigation demands which are fed to the water availability sub-model, which subsequently routes surface water through the river network, determines reservoir releases, and allocates water supply to demand. Model outputs from various processes are indicated in brown, with the granularity of model output described in brackets.

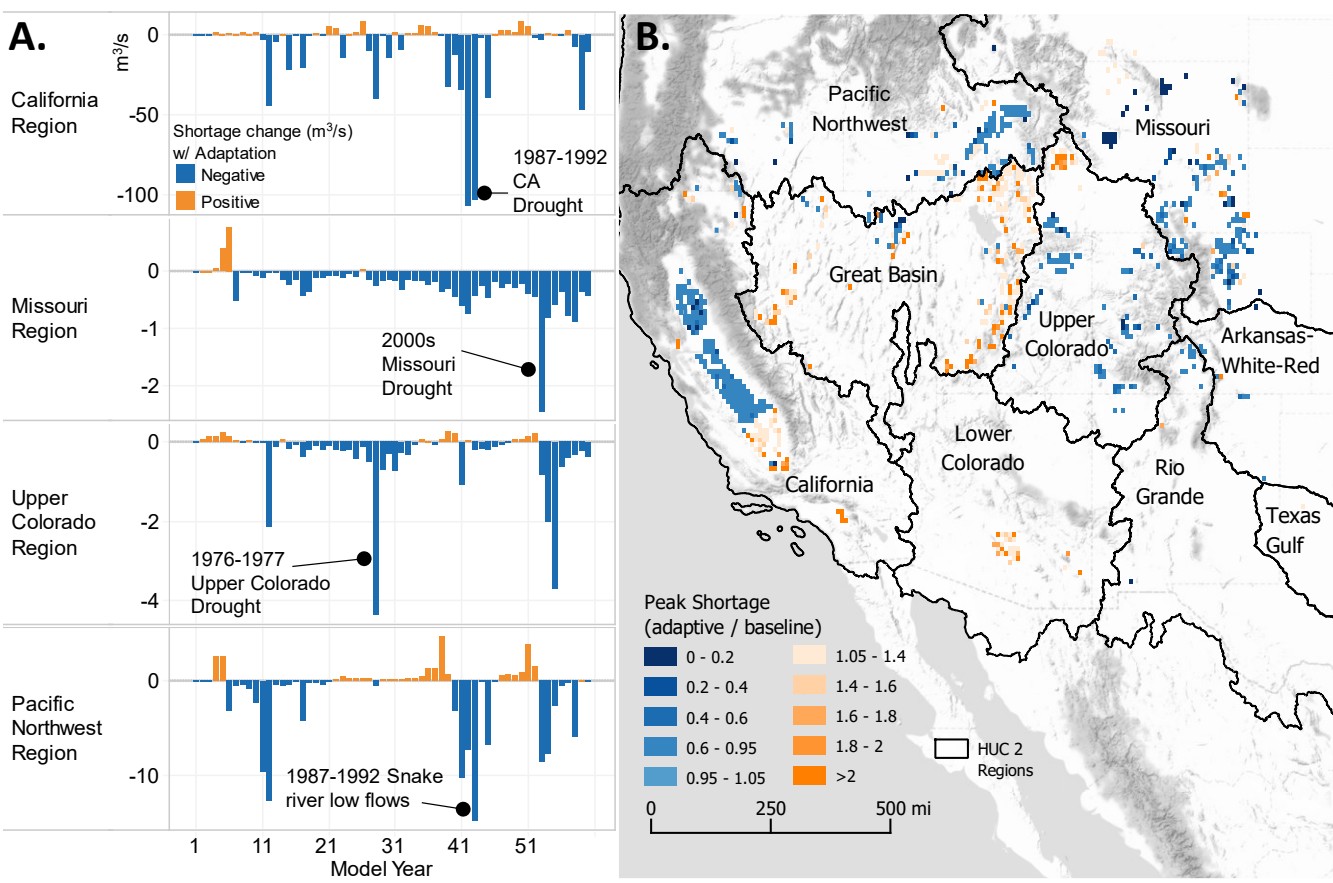

**Figure 2: Comparative water shortage results** from a hypothetical comparative model experiment mimicking 1950-2009 hydrology. (a) Shows the annual difference in water shortage between the adaptive and baseline versions of the model (annual water shortage in the adaptive run subtracted by annual water shortage in the baseline run), aggregated for four HUC 2 regions. (b) Identifies the peak annual water shortage (across all model years) for farm agents across the western United States for both the adaptive and baseline runs. The ratio of peak annual water shortage of the adaptive and baseline runs (i.e., peak annual water shortage of the adaptive divided by that of the baseline) is shown on the figure. For both (a) and (b), blue colors indicate reduced shortages with adaptation and orange colors indicate increased shortages with adaptation. Accounting for farmer cropping adaptivity results in substantial alterations in simulated water shortage across the Western United States, with a strong tendency towards mitigation of shortage. During extended periods of drought conditions in the California, Missouri, Upper Colorado, and Pacific Northwest regions, shortage is substantially reduced when taking adaptation into account (a). Peak annual shortage across the modeled time period is regularly only half of that experienced in the baseline run when accounting for adaptation (b). While shortage is curbed in most areas due to adaptation, some areas in the Great Basin, California, and Lower Colorado show an increase (indicated with orange colors).

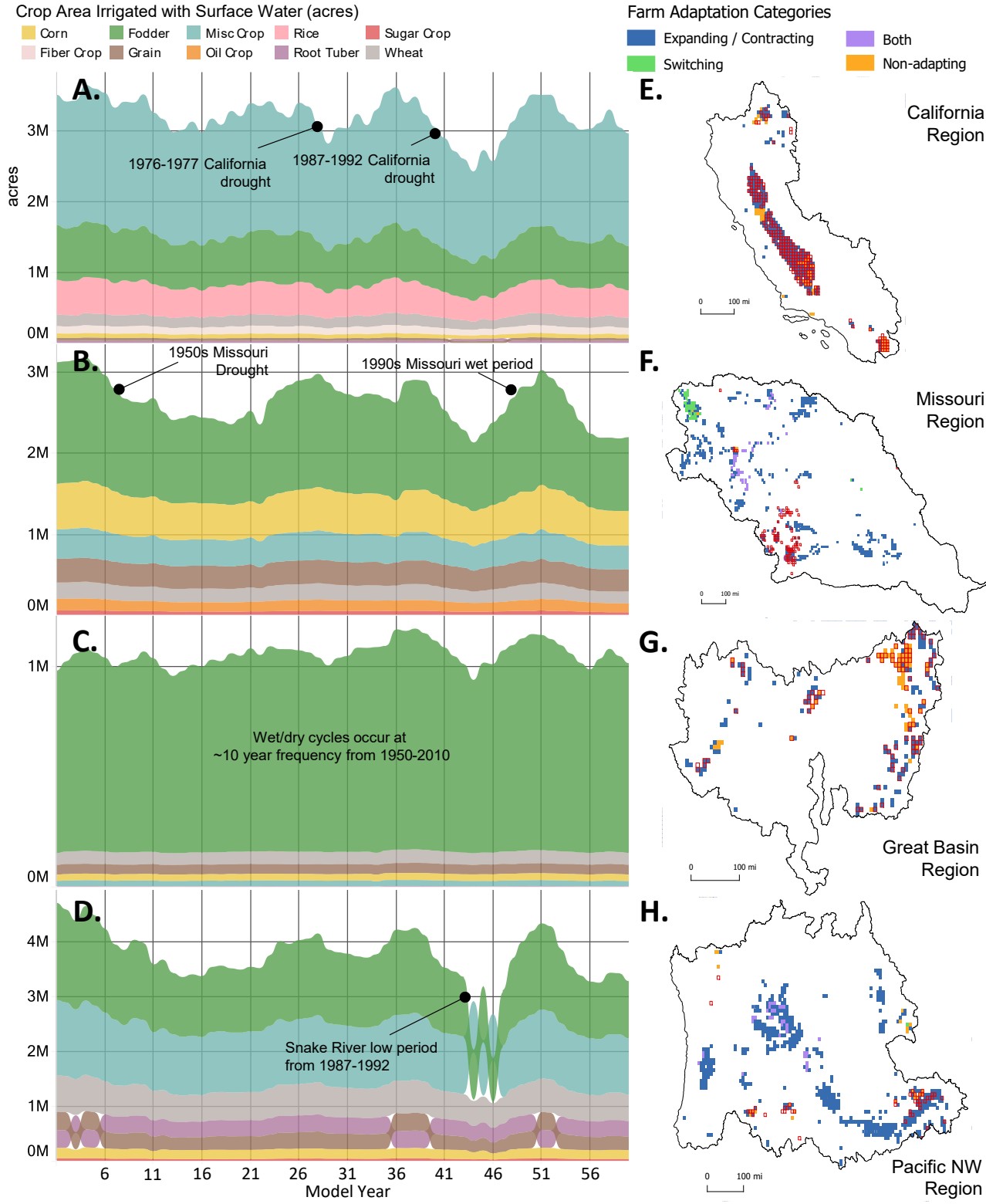

**Figure 3: Farmer cropping results** from a hypothetical comparative model experiment mimicking 1950-2009 hydrology. (a-d) Shows changes in surface-water irrigated acreages by crop for the adaptive model run, aggregated for the four HUC 2 regions of interest shown (California, Missouri, Upper Colorado, and Pacific Northwest). (e-h) classifies individual farm agents in the adaptive model run with significant irrigation into one of four categories based upon their level of cropping adaptation (looking over the entire model period): 1) "crop expansion/contraction" (in blue) if the ratio of an agent's annual minimum surface-water irrigated crop area is less than 80 percent of the annual maximum surface-water irrigated crop area, 2) " crop switching" (in green) if the predominant crop's share of the total crop makeup for any given agent (measured in terms of crop area) changes by at least 5 percent between any two years of the model run (which do not need to be consecutive), 3) "both" (in purple) if the agent satisfies both criteria 1 and 2 above, and 4) "none" (in orange) if the agent satisfies none of these criteria. Farm agents that experience significant water shortage, likewise evaluated over the entire model run, are indicated with a red outline. Expansions and contractions in irrigated cropped areas are simulated in response to major hydrological events (a-d). Farmer agents in the model largely adapt (e-h) to drought through crop contraction (blue cells with red outline), whereas crop switching (green cells) plays a less prominent role and is more prevalent in non-shortage areas. Some agents do not adapt in spite of shortage (orange cells w/ red outline).

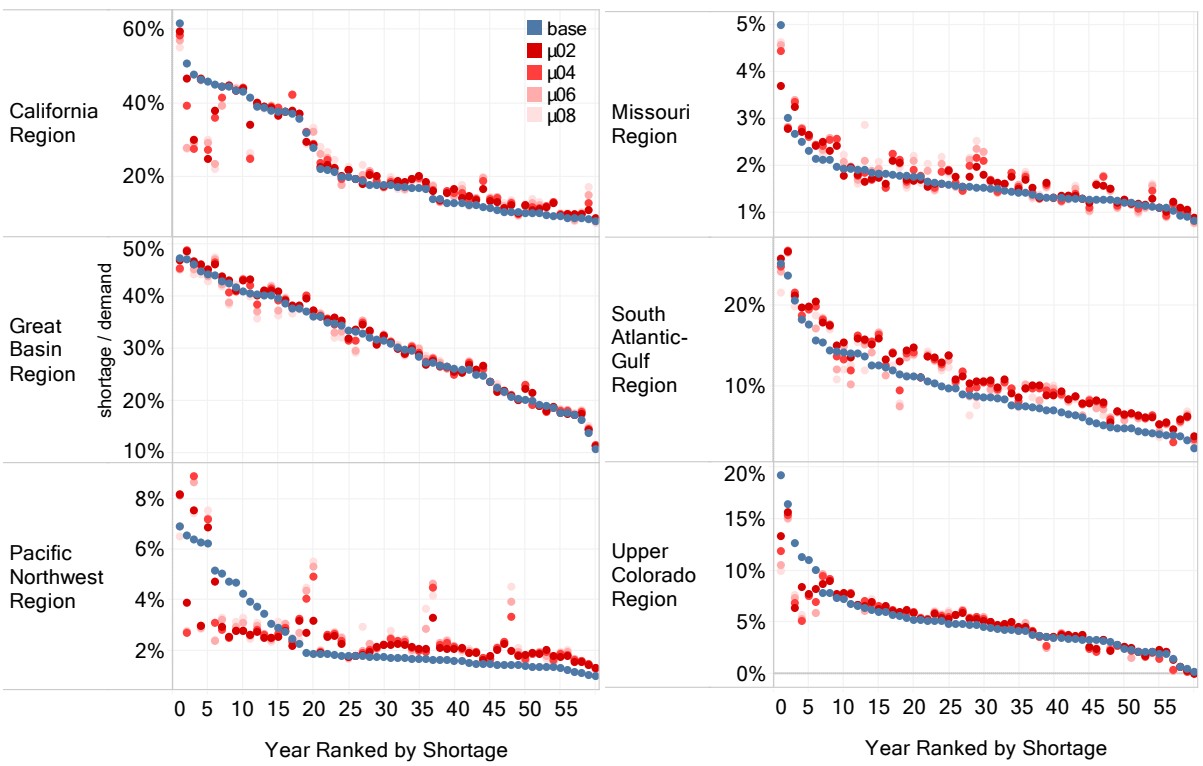

**Figure 4: Farmer agent sensitivity runs with** the annual water shortage percentage (defined as unmet water demand divided by water demand) aggregated for six HUC 2 regions and ranked by year of shortage. For each year, results from five different experiments are shown, the baseline run (indicated in blue) and four adaptive runs in which the strength of agent memory as defined through a memory decay factor μ (see methods for details) is adjusted between 0.2-0.8 (indicated in shades of red, with lighter shades of red indicating shorter agent memory, i.e., higher reactivity to more recent events). The largest reductions in water shortage due to farmer cropping adaptation occur in the highest shortage years, as most notably evidenced by the difference in the baseline and adaptive shortages for the highest shortage years (left sides of the charts) in the California, Pacific Northwest, and the Upper Colorado regions. Results are largely robust across agent memory sensitivity runs, with the general direction of difference between the baseline run (blue) and the agent adaptation runs (shaded red) usually remaining consistent (i.e., the blue dot usually lies on one end of the shaded red dots for any given region/year).

