# Peer review of "Representing Farmer Irrigated Crop Area Adaptation in a Large-Scale Hydrological Model"

_EGUsphere, 2023_

## Author Response (AR1)

We thank the reviewers for their thorough and thoughtful review of our manuscript. We have transcribed and numbered reviewer comments below. Our responses to review comments are indicated in red, with associated excerpts from our manuscript indicated in quotes with changes in blue (line numbers are in reference to the revised word doc without tracked changes).

**Reviewer #1**

The paper "Representing Farmer Irrigated Crop Area Adaptation in a Large-Scale Hydrological Model" developed an agent-based module for a large-scale hydrologic model to incorporate the adaptive crop-related decisions of farmers. I believe this is an urgent topic in the field of water resources systems analysis, the research is scientifically sound, the paper is well organized, and the results are very interesting. Nevertheless, I have the following comments for the authors to consider and potentially improve the quality of their manuscript.

We are glad the reviewer found the manuscript to be of interest, and appreciate their insightful and attentive comments. In our response below, we have copied and labeled (RX-X) individual reviewer comments followed by our response to each.

**R1-1.** First, I would suggest adding more references to the "two-way" coupling studies to smooth the logic flow. While I think this paper might be one of the first to conduct a two-way coupling of an agent-based model with a hydrologic model at the US scale, the concept itself is not brand new. There are several previous studies have already done this at the basin scale (some of the basins are fairly large such as the Mekong, Niger, and Colorado Basins). Citing these previous studies will provide readers with a better understanding of how this two-way coupling concept was developed.

We thank the reviewer for noting this and agree that additional references on two-way coupled ABM-hydrologic modeling studies would provide important context. These efforts indeed introduce and demonstrate the two-way coupling concept that serve as an inspiration for our coupling of a farm ABM with a large-scale hydrological model. Towards this end, we have added a paragraph in the introduction that points to the evolution of two-way coupled ABM-hydrologic models, primarily involving ABM couplings with SWAT, MODFLOW, and Riverware models that includes the following references:

Giuliani, M., Li, Y., Castelletti, A. and Gandolfi, C., 2016. A coupled human-natural systems analysis of irrigated agriculture under changing climate. Water Resources Research, 52(9), pp.6928-6947.

Hyun, J.Y., Huang, S.Y., Yang, Y.C.E., Tidwell, V. and Macknick, J., 2019. Using a coupled agent-based modeling approach to analyze the role of risk perception in water management decisions. Hydrology and Earth System Sciences, 23(5), pp.2261-2278.

Castilla-Rho, J.C., Rojas, R., Andersen, M.S., Holley, C. and Mariethoz, G., 2017. Social tipping points in global groundwater management. Nature Human Behaviour, 1(9), pp.640-649.

Khan, H.F., Yang, Y.C., Xie, H. and Ringler, C., 2017. A coupled modeling framework for sustainable watershed management in transboundary river basins. Hydrology and Earth System Sciences, 21(12), pp.6275-6288.

Reeves, H.W. and Zellner, M.L., 2010. Linking MODFLOW with an agent-based land-use model to support decision making. Groundwater, 48(5), pp.649-660.

Lin, C.Y., Yang, Y.E., Malek, K. and Adam, J.C., 2022. An investigation of coupled natural human systems using a two-way coupled agent-based modeling framework. Environmental Modelling & Software, 155, p.105451.

Yang, J., Yang, Y.E., Chang, J., Zhang, J. and Yao, J., 2019. Impact of dam development and climate change on hydroecological conditions and natural hazard risk in the Mekong River Basin. Journal of Hydrology, 579, p.124177.

Yang, J., Yang, Y.E., Khan, H.F., Xie, H., Ringler, C., Ogilvie, A., Seidou, O., Djibo, A.G., Van Weert, F. and Tharme, R., 2018. Quantifying the sustainability of water availability for the water-food-energy-ecosystem nexus in the Niger River Basin. Earth's future, 6(9), pp.1292-1310.

Yoon, J., Klassert, C., Selby, P., Lachaut, T., Knox, S., Avisse, N., Harou, J., Tilmant, A., Klauer, B., Mustafa, D. and Sigel, K., 2021. A coupled human–natural system analysis of freshwater security under climate and population change. Proceedings of the National Academy of Sciences, 118(14), p.e2020431118.

Klassert, C., Yoon, J., Sigel, K., Klauer, B., Talozi, S., Lachaut, T., Selby, P., Knox, S., Avisse, N., Tilmant, A. and Harou, J.J., 2023. Unexpected growth of an illegal water market. Nature Sustainability, pp.1-12.

Lines 60-69: "Our effort builds upon local and regional water modelling studies that have introduced and developed various forms of two-way coupling between agent-based models and hydrologic water systems models over recent years (Reeves and Zellner, 2010; Giuliani et al., 2016; Castilla-Rho et al., 2017; Khan et al., 2017; Yang et al., 2018; Hyun et al., 2019; Yang et al, 2019; Yoon et al., 2021; Lin et al., 2022; Klassert et al., 2023). Most of these local and regional applications have focused on capturing coupled human-hydrological interactions in an irrigated agricultural context, with some also including representation of dam operation and other water user agent categories (e.g., urban user agents). In these previous efforts, agent-based models have been integrated with models that are commonly used for case-specific representation of hydrological water systems, such as agent-based model integrations with MODFLOW (Reeves and Zellner, 2010; Yoon et al., 2021), SWAT (Khan et al., 2017), and Riverware (Hyun et al., 2019). The coupling of an agent-based model with a large-scale hydrological model distinguishes the current effort."

We also relatedly note a recent publication demonstrating a regional implementation of two-way coupled ABM-LHM integration (De Brujin et al., 2023), and have added reference to this in the revised manuscript:

Lines 55-58: "The representation of dynamic irrigated crop area adaptation in LHM frameworks for national to global scale water shortage analysis remains a major gap, though recent regional agent-based LHM implementations suggest the potential (De Bruijn, J. A. et al., 2023)."

De Bruijn, J.A., Smilovic, M., Burek, P., Guillaumot, L., Wada, Y. and Aerts, J.C., 2023. GEB v0. 1: a large-scale agent-based socio-hydrological model–simulating 10 million individual farming households in a fully distributed hydrological model. Geoscientific Model Development, 16(9), pp.2437-2454.

**R1-2.** Second, I do think more technical details should be provided in the methodology section. While the development and the connection of the two sub-models are quite clear, several technical details are only mentioned briefly. For example, the setting of groundwater supply, the real-world meaning of the two PMP coefficients, and the calculation of the adjusted perceived cost of production. To me, it is totally ok

to make some assumptions to make the model development feasible, but these details should be provided and the readers can judge themselves. Two related comments regarding methodology. 1) I think the authors should provide the ODD+D document in the supplementary material which becomes a standard in any ABM study. This will allow other ABMers to quickly understand the ABM setting. 2) I think a summary table to show the necessary data (sources, years, resolution, etc.) can help readers better understand the scope and scale of this model.

In the revised manuscript, we have added details on the treatment of groundwater supply in the framework, interpretation of the PMP coefficients, and the calculation of adjusted perceived cost of crop production. We have also incorporated a summary table on data sources (see Table 1 in the revised manuscript) that was used to initialize and parameterize the model. Finally, we have added ODD+D documentation to the supplemental materials of the manuscript, which provides further description of the technical details of the model in a standardized manner familiar to the agent-based modeling community. Revised text to the main manuscript is copied below. For purposes of space, we refer the reviewer directly to the revised manuscript materials for the data table and ODD+D documentation described above.

Lines 104-109: "As the MOSART-WM model focuses on simulating surface water availability, groundwater availability for irrigation is assumed to remain steady at the availability and cost estimated for the baseline period. For example, under an increase in surface water availability farm agents can respond by either reducing their groundwater production, or increasing their irrigated cropped areas while maintaining the same level of groundwater production. While groundwater is treated as in infinite reservoir at a static groundwater level over the simulation period, groundwater production for any given annual time step is constrained to the amount of groundwater production estimated for the calibration period. Our representation of groundwater in the model and its limitations are further addressed in the Discussion section."

Lines 157-163: "Two PMP calibration coefficients (one linear and one quadratic) are added to the profit maximization formulation. The coefficients account for the increasing marginal cost of expanding the production of any crop in a given region, due to limited local availability of crop-specific labor and capital endowments (e.g., specialized machinery, skilled workers, and farmers' knowledge), as well as heterogeneous environmental, land, and marketing conditions (Heckelei et al., 2012; Howitt et al., 2012). These increasing costs cannot be derived directly from available data on the average regional production costs, but are revealed in farmers' crop allocation decisions (Paris, 2012; Paris and Howitt, 1998). The PMP approach utilizes this by calibrating the "unobserved cost" coefficients to observed historical crop acreage data (Garnache et al., 2017)."

Garnache, C., Mérel, P., Howitt, R., & Lee, J. (2017). Calibration of shadow values in constrained optimisation models of agricultural supply. European Review of Agricultural Economics, 44(3), 363-397.

Howitt, R. E., Medellín-Azuara, J., MacEwan, D., & Lund, J. R. (2012). Calibrating disaggregate economic models of agricultural production and water management. Environmental Modelling & Software, 38, 244-258.

Howitt, R. E. (1995). Positive mathematical programming. American journal of agricultural economics, 77(2), 329-342.

Paris, Quirino (2012). Economic Foundations of Symmetric Programming. Cambridge UP: Cambridge, United Kingdom: 340–411. DOI: 10.1017/CBO9780511761782.016

Paris, Q. and Howitt, R.E. (1998). An Analysis of ill-Posed Production Problems using Maximum Entropy. American Journal of Agricultural Economics. 80(1): 124-138.

Lines 291-296: "As the baseline water demands are based on actual applied water data, these demands are assumed to account for state-to-state and crop-to-crop variation in climatology, irrigation technology, water use efficiency, and other factors that influence crop irrigation requirements. The baseline water demand and irrigation estimation is segmented by surface water and groundwater based on the source-specific estimated irrigated areas estimations described in Section 2.3.1. For groundwater specifically, this baseline groundwater irrigation estimate provides an agent-specific annual groundwater production cap (in acre-ft/year) that the agent cannot exceed for any given year of the model run."

**R1-3.** Third, I understand that this manuscript is testing a hypothetical experiment and I think it is fine for model development purposes. But I think the authors should still provide the calibration results and partially demonstrate that the model they developed (at least more or less) captures the overall trend and pattern of historical data (e.g., crop area and/or streamflow). Otherwise, it is very difficult to convince readers that the model is suitable for hypothetical experiments. One of the major criticisms of ABM is that these models are "toy models" that do not reflect reality. Since the authors use historical data to calibrate their PMP parameters, they should show the results as evidence that this is not just a toy model. Again, several two-way coupled ABM studies in recent years have already shown the calibration results to demonstrate the model's credibility. This is the most critical comment I have.

We thank the reviewer for their comment regarding the importance of calibration. We have added further analysis and results to the supplemental materials (pages 8-13) in response to the reviewer's comment which we elaborate upon below.

First, we note that the two individual sub-models have gone through their own independent calibration procedures (to a degree that is common in both their respective literatures). For the farmer sub-model, the data-driven positive mathematical programming approach and calibration is intrinsically embedded within the model development process. The identification of the PMP parameters during the first phase of model development is calibrated to observed data, such that the PMP closely reproduces observed cropping areas for the historical period (we provide new test results in the supplemental materials of the revised manuscript that verify this). As such, the PMP can be viewed as a revealed preferences methodology for specifying farm behavior, which is a commonplace economic treatment of agent behavior but not without its limitations.

With our revisions, we have also included sample local stress/sensitivity tests (e.g., changes to crop prices, water availability, etc.) to demonstrate the reasonability of farm cropping model response. These stress tests indicate reasonable behavior of the farmer model and are also included in the supplemental materials. For the VIC-MOSART-WM hydrologic modeling, attempts at calibration are detailed in various previous studies. The VIC simulation has been calibrated and is considered a benchmark for the United States Bureau of Reclamation (USBR, 2014), while the addition of the MOSART-WM component is evaluated on the basis of its performance in improving modeled river flow and reservoir storage outcomes to observations (Voisin et al., 2013; Hejazi et al., 2015).

The reviewer's comment regarding calibration of the coupled model still remains, however (hereon we specifically focus on irrigated cropped areas as our calibration performance metric, given this is the novel outcome that our model treats dynamically). While we believe a calibration over the full historical period of interest is outside the scope of the effort for reasons we will elaborate below, we can nonetheless assess the reasonability and plausibility of model behavior by comparing irrigated crop area outcomes from our simulations with known cropping response during isolated periods of drought (over which we assume that

non-hydrologic external factors are relatively stable). This can provide us with an indication of how our adaptive model is performing relative to the alternative (no cropping adaptation).

Following the reviewer's comment, we have conducted such a comparison for several states in the Western U.S. for the early 2000s period, during which much of the region was in drought. Over these isolated periods (over which we assume, albeit imperfectly, that non-hydrological conditions are relatively steady), our model generally reproduces changes in irrigated crop areas that are commensurate with observed changes (and in all but a couple of cases gets the direction of cropping change correct). This can be readily contrasted with a non-adaptive model, in which the irrigated crop area changes would all be zero based on the fundamental design of the model. We believe results such as these indicate that our approach offers a clear improvement over the non-adaptive cropping assumptions embedded in a traditional LHM as assessed against real-world data. Results from this new model plausibility exercise have been included in the revised supplementary materials.

Beyond the calibration of individual model components and the plausibility evaluation described above, a full time-series calibration of irrigated cropped areas using the coupled model presents several challenges. While our study focuses on and endogenizes the influence of water availability on irrigated cropping decisions, there are many factors outside of water availability that influence irrigated cropping patterns. Other factors such as crop prices, crop production costs, areas equipped with irrigation, the existence and capacity of dams/reservoirs, subsides on crop production, and inter-basin water transfers are among these external influences. To conduct a coherent historical calibration, we would need to account for all of these time-varying factors alongside our dynamic treatment of hydrologically-driven water availability, one that is extensive over CONUS and conducted for a model time period of reasonable length to capture adequate variability of the coupled system (in our view, at least ~50 years), a major and likely unprecedented undertaking which we consider outside the scope of the current analysis. Similarly, a comprehensive calibration of our model would require a reliable record of observed Surface Water/Groundwater irrigated cropped areas over the historical period of calibration, for which we are unaware of an existing dataset. For these reasons, we have taken the approach to strongly emphasize that the effort is a hypothetical analysis (rather than a historical reconstruction) in our initial manuscript, as the reviewer notes.

We have included the full results of our new PMP calibration validation, PMP stress tests, and model plausibility evaluation exercise in the supplemental materials of the revised manuscript. Due to the length of these materials, we refer the reviewer directly to pages 8-13 of the revised supplemental materials. Below, we highlight a key paragraph that explains the challenges of a full model calibration, including new text added to the revised manuscript that describes the stress and plausibility evaluations and points readers to the new content in the supplemental materials.

Lines 424-433: "We note that the study is designed as a hypothetical experiment to evaluate the influence of farmer cropping adaptation on modeled water shortage outcomes rather than an attempt at a historical reconstruction of actual cropping and water use patterns, as other non-hydrological influences such as crop prices, areas equipped with irrigation, and crop-specific irrigation requirements remain static over the model run. The hypothetical experiment is rather designed to identify the potential cropping adaptation response to hydrologically-driven changes in water availability (using the 1950-2009 record as a reasonable window of hydrological variability), holding all other influences constant. As such, a detailed comparison of model results against observed data is not applicable, though we evaluate the plausibility and reasonability of model results by stress testing the farmer agent model as well as comparing modeled land use changes with observed land use changes over an isolated period of drought in the Western United States to determine whether modeled crop adaptations are commensurate with historical observations (see supplemental materials)."

Reclamation, U.S., 2014. Downscaled CMIP3 and CMIP5 climate and hydrology projections: Release of hydrology projections, comparison with preceding information, and summary of user needs. Denver, CO: US Department of the Interior, Bureau of Reclamation, Technical Services Center.

Voisin, N., Li, H., Ward, D., Huang, M., Wigmosta, M. and Leung, L.R., 2013. On an improved sub-regional water resources management representation for integration into earth system models. Hydrology and Earth System Sciences, 17(9), pp.3605-3622.

Hejazi, M.I., Voisin, N., Liu, L., Bramer, L.M., Fortin, D.C., Hathaway, J.E., Huang, M., Kyle, P., Leung, L.R., Li, H.Y. and Liu, Y., 2015. 21st century United States emissions mitigation could increase water stress more than the climate change it is mitigating. Proceedings of the National Academy of Sciences, 112(34), pp.10635-10640.

I have some minor comments below to help the authors to improve the readability.

**R1-4**. Line 85: Does this mean groundwater is an "infinite underground reservoir?" It is ok for this assumption but need to make it clear to the readers.

Yes, groundwater is treated as an infinite reservoir over the modeled time horizon, however production capacity and costs are fixed to the historical period (i.e., a farmer agent cannot simply produce groundwater free of cost or volume limitations in the face of surface water shortage). We have clarified this in the revised manuscript.

Lines 106-109: "While groundwater is treated as in infinite reservoir at a static groundwater level over the simulation period, groundwater production for any given annual time step is constrained to the amount of groundwater production estimated for the calibration period. Our representation of groundwater in the model and its limitations are further addressed in the Discussion section."

**R1-5**. Line 97: This is interesting. Is there a reason why this 50 km threshold? How sensitive is this threshold?

The 50 km threshold represents a reasonable estimate of a distance cutoff for most diversions. In the initial implementation of this threshold-based approach, Biemans (2011) performed a sensitivity analysis on this threshold value, concluding that an increase to 100 km would increase demand by 4 percent while a decrease in the buffer to 25 km would decrease demand by 18 percent. The selection of 50 km is also chosen due to computational tradeoffs. As the buffer increases, additional agents/cells have access to any given reservoir, increasing the computational requirement for the reservoir water allocation algorithm. We have made the following revision to the manuscript:

Lines 123-126: "The 50 km threshold represents a reasonable estimate of a distance cutoff for most diversions (Biemans et al., 2011), while also aimed at reducing computational expense (as the buffer increases, additional agents/cells have access to any given reservoir, increasing the computational requirement for the reservoir water allocation algorithm)."

Biemans, H., Haddeland, I., Kabat, P., Ludwig, F., Hutjes, R. W. A., Heinke, J., von Bloh, W., and Gerten, D. (2011), Impact of reservoirs on river discharge and irrigation water supply during the 20th century, Water Resour. Res., 47, W03509, doi:10.1029/2009WR008929.n procedure.

**R1-6**. Line 103: I assume the hydrological proxy during the calibration period is like a long-term average?

Yes, the hydrological proxy during the calibration period can be viewed as a long-term average of a hydrologic state (either a surface water reservoir level or a runoff value). Agents look to this hydrologic state to formulate their expectation of water availability. In simulation mode, the dynamically simulated hydrological proxy results in changes to agents' expectation of water availability as the model advances in time. We have made the following clarification to the manuscript:

Lines 129-132: "Determine an adjusted surface water demand (Demadj) by dividing the updated hydrological proxy by the hydrological proxy during the calibration period (Hb) and multiplying by the surface water demand during the calibration period (Demb). The hydrological proxy during the calibration period can be viewed as a long-term average of the hydrological state identified in step 1 above."

**R1-7.** Line 173: I assume "Data Sources and Processing" mean Section 2.3? Because there is no sub-section with this title.

Yes, we have made the following correction to the manuscript:

Line 205: "The data sources and processing for the agent calibration are described further in the 'Farm Data for PMP Calibration' sub-section below."

**R1-8.** Line 188: Is there a specific reason why this period: 2010-2013 is used?

We select the 2010-2013 period due to both data availability as well as historic drought conditions: 1) the Cropland Data Layer, a critical input for a our data workflow, is only available starting in 2008, 2) the start of the 2010s were a period in which historic drought over the United States was relatively low as a baseline, and 3) the USDA Farm and Ranch Irrigation Survey is only available in 2013. We combine these data sources together (CDL starting in 2010) and consider them a historic representation of 2010-2013 conditions. We have added the following explanation/justification to the revised manuscript:

Lines 221-231: "For our purposes, we assume that these various data sources are an averaged representation of the 2010-2013 historical period, though we recognize that the data sets are drawn from different years and that conditions may be variable within this time period. We select the 2010-2013 period due to both data availability as well as historic drought conditions: 1) the Cropland Data Layer (CDL), a critical input for a our data workflow, is only available starting in 2008, 2) the start of the 2010s were a period in which historic drought over the United States was relatively low as a baseline (i.e., we are not calibrating to an atypical period of water shortage), and 3) the USDA Farm and Ranch Irrigation Survey is only available in 2013. We combine these data sources together (CDL starting in 2010) and consider them a historic representation of 2010-2013 conditions. Exploring the sensitivity of PMP parameters and model behaviour to the choice and uncertainty of these input datasets is an important future research direction. For example, the PMP could be recalibrated based on different years in which the CDL and USDA Farm and Ranch Irrigation Survey is available, assessing the sensitivity of the calibrated PMP coefficients against those generated with the data used for this particular study."

**R1-9.** Line 233: I assume farm-level optimization is agent level in Section 2.2?

Correct, we have clarified to indicate "agent-level" in our revisions:

Line 273: "For these cells, we set the available land area constraints in the agent-level optimization (see above) to this total observed crop area such that the PMP can reproduce the calculated total land areas but are unable to exceed them in simulation mode."

**R1-10**. Line 246: I think the calculation is Water Demand = crop area * irrigation requirement. In the ABM, when you adjust water demand, did you change both crop area and irrigation requirement? Or do you only change the crop area?

We only change crop area. The irrigation requirement (e.g., the depth of irrigation water required per unit land area of crop planted) is assumed static over our model run, i.e., we do not consider the impacts of climate change on the irrigation requirement for our analysis. We have clarified this in our revisions.

Lines 287-291: "In an adaptive model simulation, agents are initialized with their baseline water demands, and subsequently adjust their water demands as they adapt to changing water availability conditions as the model steps through time. In our formulation, only cropped areas are assumed to change over time (the irrigation requirement, i.e., the depth of irrigation water required per unit land area of crop planted is assumed to remain static over the model run)."

**R1-11**. Line 252 and 265: Are these assumed to be the same throughout the simulation period? It is fine if that is the case, but should be clarified.

Yes, all unit prices and costs are assumed to be the same throughout the simulation period. We have clarified this in the revised manuscript.

Lines 309-311: "For empirical reasons we remove the USDA estimates of unpaid labor costs and imputed opportunity costs of land, as the PMP calibration terms are better able to capture the relevance and heterogeneity of these non-monetary production costs. Crop prices and production costs are static over the simulation period."

**R1-12**. Line 256: Do you mean Table S1?

Yes, we have made this correction:

Lines 302-304: "Similar to the USDA irrigation survey, ERS crop categories are mapped to more general crop categories to enhance compatibility with other global models (as detailed in Table S1 in the supplemental materials)"

**R1-13**. Line 286: Is this a typo for PerceivedCost?

Yes, we have made this correction:

Line 334: "$Profit_{n,c} = (Yield_{n,c} * Price_{n,c}) - PerceivedCost_{n,c}$"

**R1-14**. Line 292: Can you provide an equation?

Yes, the equation below has been added to the manuscript:

Lines 343-345:

"$PerceivedCostAdj_{n,c} = (Yield_{n,c} * Price_{n,c}) - Profit_{n,c}$   when   $Profit_{n,c} \geq .10 * Yield_{n,c} * Price_{n,c}$

$PerceivedCostAdj_{n,c} = (.90 * Yield_{n,c} * Price_{n,c})$   when   $Profit_{n,c} < .10 * Yield_{n,c} * Price_{n,c}$"

**R1-15**. Line 312: I think you might need to mention what is VIC first. Also, is irrigation water availability = streamflow in each grid?

Noted. Considering the reviewer's comment below as well (R1-16), we will merge sub-section 2.3.6 into 2.4 after the introduction of the VIC-MOSART-WM descriptions in the following sub-sections, since 2.3.6 pertains to both the agent model and the hydrologic model. "Irrigation water availability" refers to both local streamflow in a coincident grid cell to the farmer, as well as water made available via allocation from a surface water reservoir. The combined/shortened section in the revised manuscript now reads:

Lines 360-391: "Surface water availability that feeds into the farm agents' cropping and irrigation decisions is dynamically provided by MOSART-WM. MOSART-WM is a spatially distributed large-scale water management model consisting of a physically based river-routing model (MOSART, Li et al., 2013) coupled with a generalized water-management model (WM, Voisin et al. 2013b) for seasonal to long term studies. MOSART-WM takes surface runoff generation input from an external hydrological model, commonly the Variable Infiltration Capacity (VIC) model in previous applications. In the river-routing component, daily surface runoff is an input that is first routed across hill slopes and then discharged into a tributary subnetwork within each grid cell before entering the main channel for transport across grid cells. WM has two components – reservoir operations which influence the seasonality of flow and river storage, and water supply management which allocates supply from reservoirs to spatially distributed demands across grid cells.

Our CONUS set up of MOSART-WM model includes all (1,848) reservoirs with a storage capacity larger than 10 million m3, i.e. focusing on reservoirs that most influence river discharge. The reservoir database and locations are obtained from the Global Reservoir and Dam Database (GranD) reservoir database (Lehner et al., 2011). For daily reservoir storage and release operations, MOSART-WM adopts generic operating rules that mimic monthly release and storage patterns based on the objective of the reservoir (e.g., flood control, irrigation, etc.), its physical characteristics (storage) and monthly climatologies of inflow and demand, and follows daily constraints for minimum environmental flow, and minimum and maximum storage volumes. The reservoir model builds upon generic operating rules introduced in Biemans et al. (2011) and Hanasaki et al. (2006) and improved upon for multi-objective operations by Voisin et al. (2013b).

During the calibration phase of the PMP model, we assume that agent's water availability constraints are non-binding (i.e., the irrigation water required to produce the observed surface-water irrigated crop area estimates were available during the calibration period). To account for potential inconsistencies between the estimated surface water irrigation demand (as estimated via the data sources and procedures described above) and VIC-MOSART-WM simulated irrigation water availability for the calibration period, we then apply a bias correction factor to the crop irrigation requirements on a cell-by-cell basis such that total irrigation demand matches VIC-MOSART-WM simulated water availability for the historical calibration period. Such a treatment attempts to reconcile potential inconsistencies between estimated irrigation requirement calculated from the data assimilation process described throughout the sub-sections above and actual irrigation water availability modeled by VIC-MOSART-WM for the baseline period. The additive cell-specific bias correction factor is subsequently applied in simulation mode for each time period. This approach addresses potential biases in VIC-MOSART-WM's estimates of irrigation water availability for baseline conditions, and assumes these biases are maintained in the ABM simulations that depart from baseline conditions while relying on modeled results to estimate changes in water availability relative to the baseline condition."

**R1-16.** Line 322 (Section 2.4): Don't really think you need one page of these backgrounds since these are already published. So maybe merge this section with 2.3.6 to smooth the logic flow? Otherwise, a sudden mention of simulated irrigation water availability is a bit logical jump.

Noted. We have merged with Section 2.4 (see response to previous comment) and significantly shortened the background material on VIC-MOSART-WM.

**R1-17.** Line 371: I think an equation to show how you calculate water shortage is still necessary.

Yes, we have included an equation with the revised manuscript:

Line 407: "$WaterShortage = WaterDemand - LocalWaterSupply - ReservoirWaterSupply$"

**R1-18.** Line 393: I think some text in the caption of Fig 2 should be moved (or copied) to the main text like how you calculate the blue and orange bars and dots.

We have included this information (calculation of the colored bars and dots) in the main text as well, while also keeping it in the caption so readers have access to the information in both locations.

Lines 436-443: "Accounting for farmer irrigated crop area adaptation substantially alleviates simulated annual water shortages, especially during periods of severe regional drought (Fig. 2, see Fig. S2 for monthly details). In Figure 2, we show the annual difference in water shortage between the adaptive and baseline versions of the model (annual water shortage in the adaptive run subtracted by annual water shortage in the baseline run), aggregated for four HUC 2 regions. We also identify the peak annual water shortage (across all model years) for farm agents across the western United States for both the adaptive and baseline runs, with the ratio of peak annual water shortage of the adaptive and baseline runs (i.e., peak annual water shortage of the adaptive divided by that of the baseline) shown on the Fig. 2. Blue colors indicate reduced shortages with adaptation and orange colors indicate increased shortages with adaptation."

**R1-19.** Line 394: Is it a typo for Fig2b?

Yes, we have corrected this typo in the revised manuscript:

Line 444: "Water shortage alleviation due to adaptation is especially prevalent across the Western United States (Fig. 2b)…"

**R1-20.** Line 426 and 444: I think you need to at least show the Eastern US results in the supplementary material because you do emphasize in your abstract and introduction that this is a CONUS study. But currently, there are no results showing this scale.

Thank you for calling attention to this. We have added complete CONUS results (i.e., a full CONUS map of Figure 2b and Figures 2e-h) as Figure S3 and S4 in the Supplemental Materials. We also reference these new figures in the revised text of the main manuscript:

Lines 477-479: "Across the eastern United States (east of the Mississippi River), water shortage differences between the adaptive run and the baseline run are subdued, with significant changes isolated to southern pockets of the Texas Gulf, Lower Mississippi, and South-Atlantic Gulf regions (see Fig. S3 in the Supplemental Materials for a full CONUS map)."

Lines 494-496: "Model results indicate that irrigated crop adaptation activity is heavily concentrated in the Western United States, with relatively subdued activity in the Eastern United States outside of the Lower Mississippi and South Atlantic-Gulf regions (see Fig. S4 in the Supplemental Materials for a full CONUS map)."

**R1-21**. Line 435: It is a bit unclear how you calculate Fig 3e-h? Is it a long-term average? Counting every model year?

The nature of the classification is provided in lines 435-441, copied below. The classification is conducted looking over every year of the model run. For example, an agent is classified as "crop expansion/contraction" if the stated criteria is satisfied, considering every model year of the 60-year model period.

"…while Figure 3e-h assigns farms to crop adaptation categories based on the amount of crop adaptation simulated over the model period. For the latter, agents are assigned to one of four categories depending upon the level of crop adaptation activity: 1) "crop expansion/contraction" if the ratio of an agent's annual minimum surface-water irrigated crop area is less than 80 percent of the annual maximum surface-water irrigated crop area, 2) " crop switching" if the predominant crop's share of the total crop makeup for any given agent (measured in terms of crop area) changes by at least 5 percent between any two years of the model run (which do not need to be consecutive), 3) "both" if the agent satisfies both criteria 1 and 2 above, and 4) "none" if the agent satisfies none of these criteria."

We have clarified the nature of the classification in our revisions. We have also included a brief explanation in the caption of Fig 3 for clarity.

Lines 487-489: "For the latter, agents are assigned to one of four categories depending upon the level of crop adaptation activity, conducted for each agent considering every year of model output: 1) "crop expansion/contraction" if the ratio.."

Lines 885-893: "Figure 3: Farmer cropping results from a hypothetical comparative model experiment mimicking 1950-2009 hydrology. (a-d) Shows changes in surface-water irrigated acreages by crop for the adaptive model run, aggregated for the four HUC 2 regions of interest shown (California, Missouri, Upper Colorado, and Pacific Northwest). (e-h) classifies individual farm agents in the adaptive model run with significant irrigation into one of four categories based upon their level of cropping adaptation (looking over the entire model period): 1) "crop expansion/contraction" (in blue) if the ratio of an agent's annual minimum surface-water irrigated crop area is less than 80 percent of the annual maximum surface-water irrigated crop area, 2) " crop switching" (in green) if the predominant crop's share of the total crop makeup for any given agent (measured in terms of crop area) changes by at least 5 percent between any two years of the model run (which do not need to be consecutive), 3) "both" (in purple) if the agent satisfies both criteria 1 and 2 above, and 4) "none" (in orange) if the agent satisfies none of these criteria."

**R1-22**. Line 470: I don't see the results of agricultural profits. Are you showing any figures or tables?

We have not included figures/tables of agricultural profits in the main manuscript text to keep it at a reasonable length and to maintain focus on the primary outcome of our model (changes in irrigated crop area and ensuing water shortage). We also note a nuance – our model outputs expected agricultural profit rather than actual agricultural profit, as impacts of water shortage on crop yields and ensuing profitability are not calculated by the model. However, we believe expected agricultural profit results are still useful and insightful, and have included a figure showing expected agricultural profits for each HUC 2 region in

the Supplementary Materials (Figure S5). We also include a reference to this new material in the main manuscript text:

Lines 531-532: "Results showing expected agricultural profits for all HUC 2 regions and model years are included in the Supplemental Materials."

**R1-23**. Line 545: I think there is another limitation worth mentioning which is social norm effects. Farmers' behaviors are heavily affected by their social networks (neighbors, friends, etc.). There are ABM out there showing this already and can be considered in the future.

We thank the reviewer for calling attention to this important social phenomenon and agree this is a limitation of the current approach (as well as an opportunity for improvement). We do additionally note that the large-scale nature of the current effort presents an interesting conceptual challenge for representation of social norms. A single representative farm in our case can represent hundreds of neighboring farms in reality. Implicitly, this aggregation assumes that all farms accounted for by a representative farm follow the same decision model and set of norms.

We will call attention to this limitation and challenge in the revised manuscript, and also provide reference to key previous studies with explicit treatment of social norms noted below:

Lines 621-627: "Recent efforts have further explored the effects of social norms on the influence of farmer behavior, for example evaluating the influence of social norms on farmer forecasts (Hu et al., 2006). Such considerations have been operationalized in coupled human-water system models, such as representing social norms on compliance in irrigated agricultural groundwater systems (Castilla-Rho et al., 2017) and beliefs about water supply conditions in a reservoir-controlled river system (Lin et al., 2022). The incorporation of social norms effects into ABM integration with LHMs represents another future avenue of research, though considerations of aggregation and scale (e.g., highly aggregated representative agents in LHMs) pose additional conceptual challenges relative to more highly-resolved applications."

Hu, Q., Zillig, L.M.P., Lynne, G.D., Tomkins, A.J., Waltman, W.J., Hayes, M.J., Hubbard, K.G., Artikov, I., Hoffman, S.J. and Wilhite, D.A., 2006. Understanding farmers' forecast use from their beliefs, values, social norms, and perceived obstacles. Journal of applied meteorology and climatology, 45(9), pp.1190-1201.

Castilla-Rho, J.C., Rojas, R., Andersen, M.S., Holley, C. and Mariethoz, G., 2017. Social tipping points in global groundwater management. Nature Human Behaviour, 1(9), pp.640-649.

Lin, C.Y., Yang, Y.E., Malek, K. and Adam, J.C., 2022. An investigation of coupled natural human systems using a two-way coupled agent-based modeling framework. Environmental Modelling & Software, 155, p.105451.

**Reviewer #2**

The study introduces an agent-based module into a hydrological model to integrate the dynamic decision-making processes of farmers regarding crop-related choices. The research demonstrates strong scientific rigor, the paper maintains a well-structured organization, and the findings are notably captivating. I would like to offer the following constructive feedback to the authors, which could potentially enhance the overall quality of their manuscript.

We are glad the reviewer found the manuscript to be interesting and are thankful for their insightful comments. In our response, we have copied and labeled (RX-X) individual reviewer comments followed by our response to each.

**R2-1**. My main comment is regarding the use of surface water only. Most irrigation in the USA is from groundwater. How are the results impacted by this assumption? Groundwater pumping can be considered an adaptation strategy when surface water runs out.

The reviewer rightfully notes the importance of groundwater for crop irrigation in the United States. We firstly clarify that groundwater is accounted for in our formulation of farmer agent behavior; however, the hydrologic state of the groundwater system (i.e., groundwater levels, wellfield capacities, and groundwater quality) is assumed to remain static over the course of the model run. In the farmer agent formulation, agents can access groundwater for crop irrigation, but this production comes at a cost to the farmer (per volumetric unit of groundwater produced) and is limited by a capacity of groundwater production (that is assumed to be the groundwater production for irrigation estimated via observations during the calibration period). The latter is perhaps the most significant limitation of groundwater treatment in our effort. In our revisions, we will clarify the treatment of groundwater in both the farmer agent formulation as well as on the hydrologic side of the model (see also response to Reviewer #1s R1-2 comment).

In the current discussion section, we note implications of our treatment of groundwater on our study conclusion. Namely we note that the "ability of farms to increase groundwater extraction in response to surface water shortage, as well as changes in the availability and cost of groundwater (e.g., due to depletion of groundwater in stressed aquifers) are not currently represented in our modeling framework. These potential responses may either mute the simulated water shortage changes (e.g., instances in which farmers increase groundwater pumping to accommodate surface water shortage) or heighten them (e.g., instances in which increasing groundwater depletion results in even higher shortages and ensuing adaptive responses)." It is difficult to hypothesize a consistent implication given these considerations. In some cases, farms may indeed compensate surface water shortage by simply pumping more groundwater where physically available and economically viable, though this would still impact their agricultural profitability. In other cases, groundwater production for irrigation may already be at its limits in terms of production capacity, cost, and/or water quality and further dwindling as aquifers are increasingly stressed. In these cases, declines in groundwater accessibility may interact with surface water availability in unexpected ways. We will elaborate upon this in our manuscript revisions.

Given these considerations, we also note ongoing work that is outside the scope of the current manuscript, but is focused on the addition of a CONUS-scale groundwater response simulator into the MOSART-WM-ABM framework introduced in this manuscript. This improvement, which is still in its early stages, will allow us to address the groundwater dynamics that the reviewer calls attention to. We will note this ongoing effort as a future research direction in our revised manuscript. Revisions to the manuscript include:

Lines 104-109: "As the MOSART-WM model focuses on simulating surface water availability, groundwater availability for irrigation is assumed to remain steady at the availability and cost estimated for the baseline period. For example, under an increase in surface water availability farm agents can respond by either reducing their groundwater production, or increasing their irrigated cropped areas while maintaining the same level of groundwater production. While groundwater is treated as in infinite reservoir at a static groundwater level over the simulation period, groundwater production for any given annual time step is constrained to the amount of groundwater production estimated for the calibration

period. Our representation of groundwater in the model and its limitations are further addressed in the Discussion section."

Lines 598-612: "We further note additional key limitations of the current modeling framework. While we represent groundwater production for irrigation in the modelling approach, the availability and cost of groundwater production is held fixed to baseline calibration conditions, a limitation of our analysis. The ability of farms to increase groundwater extraction in response to surface water shortage, as well as changes in the availability and cost of groundwater (e.g., due to depletion of groundwater in stressed aquifers) are not currently represented in our modeling framework. These potential responses may either mute the simulated water shortage changes (e.g., instances in which farmers increase groundwater pumping to accommodate surface water shortage) or heighten them (e.g., instances in which increasing groundwater depletion results in even higher shortages and ensuing adaptive responses). It is challenging to suggest a consistent implication of improving groundwater dynamics in the model; in some cases, farms may compensate surface water shortage by pumping more groundwater where this is physically and economically viable, while in other cases groundwater depletion may lead to reductions in production (due to changes in capacity, cost, or quality) that intertwine with surface water availability changes in a more complex manner. Evolution of such changes over time given regional hydrological and agronomic differences remains a major question. The incorporation of a dynamic sub-model for groundwater aquifer response to surface water conditions and groundwater pumping thus presents an important avenue for future research and model improvement, one that is anticipated as a future update to the initial version of the model introduced in this work."

**R2-2**. Line 30: Please quantify how much water is consumed and withdrawn by irrigation in the world and the USA.

We agree that this information would provide helpful context for the manuscript, and have included such estimates in the introduction of our revised manuscript:

Lines 32-36: "Modelling studies estimate that ~2,700 ± 540 km3 of water is withdrawn globally each year for irrigation and ~1,200 ± 99 km3 of that water consumed (McDermid et al., 2023), though such estimates of global irrigation are prone to considerable uncertainty (Puy et al., 2022). Based on a country-specific estimate for the United States (Dieter et al,, 2018), irrigation water withdrawals were estimated at 163 km3 in 2015, with consumptive use for irrigation estimated at 101 km3"

McDermid, S., Nocco, M., Lawston-Parker, P., Keune, J., Pokhrel, Y., Jain, M., Jägermeyr, J., Brocca, L., Massari, C., Jones, A.D. and Vahmani, P., 2023. Irrigation in the Earth system. Nature Reviews Earth & Environment, pp.1-19.

Puy, A., Sheikholeslami, R., Gupta, H.V., Hall, J.W., Lankford, B., Lo Piano, S., Meier, J., Pappenberger, F., Porporato, A., Vico, G. and Saltelli, A., 2022. The delusive accuracy of global irrigation water withdrawal estimates. Nature communications, 13(1), p.3183.

Dieter, C.A., Maupin, M.A., Caldwell, R.R., Harris, M.A., Ivahnenko, T.I., Lovelace, J.K., Barber, N.L. and Linsey, K.S., 2018. Estimated use of water in the United States in 2015 (No. 1441). US Geological Survey.

**R2-3**. Line 32-33: There is extensive work on irrigation expansion done by Rosa and colleagues. Elliot et al., 2014 did not quantify irrigation expansion, but impacts of climate change on current irrigation.

https://www.science.org/doi/full/10.1126/sciadv.aaz6031

https://iopscience.iop.org/article/10.1088/1748-9326/aadeef/meta

https://www.pnas.org/doi/abs/10.1073/pnas.2017796117

https://iopscience.iop.org/article/10.1088/1748-9326/ac7408/meta

We thank the reviewer for their clarification of Elliot et al., 2014 and agree with their assessment. Upon revisiting the manuscript, we realized that our sentence was phrased in a confusing manner that could be prone to misinterpretation. We have rephrased the sentence to clarify that Elliot et al., 2014 evaluates the impacts of climate change on current irrigation.

We also thank the reviewer for pointing our way to the insightful studies by Rosa and colleagues that attempt to quantify potential irrigation intensification and expansion. In our revised manuscript, we have included reference to the four publications the reviewer notes, which improves the context, motivation, and relevance of our efforts.

Lines 36-40: "While climate change threatens the availability of water supply to sustain current irrigation practices (Elliot et al., 2014) and inter-sectoral competition for water resources may also limit irrigation potential (Rosegrant et al., 2002), opportunities for sustainable irrigation expansion have also been identified to enhance food security under both current and future climatic conditions (Rosa et al., 2018; Rosa et al., 2020a; Rosa et al., 2020b; Rosa, 2022)"

**R2-4**. Line 35: Recent literature that quantified the contribution of dam-based water storage on irrigation: https://www.pnas.org/doi/abs/10.1073/pnas.2214291119

Thank you for calling attention to this important study. While we are under the impression that study does not particularly fit into the list of references included in line 35, which are all studies associated with large-scale hydrologic models of a distinct lineage, we nonetheless think the findings of the study are important to highlight. We have included a separate sentence indicating its relevance to our work in our revisions.

Lines 40-43: "Dams have been noted to play a unique and critical role in realizing this future irrigation potential (Schmitt et al., 2022). The role of dams on the enablement of irrigation has been a particular focus of several large-scale modeling analyses (Hanasaki et al., 2006; Fekete et al., 2010; Biemans et al., 2011; Voisin et al., 2013a; Haddeland et al., 2014)."